# Direct comparison of spatial transcriptional heterogeneity across diverse *Bacillus subtilis* biofilm communities

Yasmine Dergham [1,2], Dominique Le Coq[1,3], Pierre Nicolas [4], Elena Bidnenko[1], Sandra Dérozier[4], Maxime Deforet[5], Eugénie Huillet[1], Pilar Sanchez-Vizuete[1], Julien Deschamps[1], Kassem Hamze [2] ✉ & Romain Briandet [1] ✉

*Bacillus subtilis* can form various types of spatially organised communities on surfaces, such as colonies, pellicles and submerged biofilms. These communities share similarities and differences, and phenotypic heterogeneity has been reported for each type of community. Here, we studied spatial transcriptional heterogeneity across the three types of surface-associated communities. Using RNA-seq analysis of different regions or populations for each community type, we identified genes that are specifically expressed within each selected population. We constructed fluorescent transcriptional fusions for 17 of these genes, and observed their expression in submerged biofilms using time-lapse confocal laser scanning microscopy (CLSM). We found mosaic expression patterns for some genes; in particular, we observed spatially segregated cells displaying opposite regulation of carbon metabolism genes (*gapA* and *gapB*), indicative of distinct glycolytic or gluconeogenic regimes coexisting in the same biofilm region. Overall, our study provides a direct comparison of spatial transcriptional heterogeneity, at different scales, for the three main models of *B. subtilis* surface-associated communities.

Spatially organised communities, such as biofilms, are embedded in a self-produced extracellular matrix and exhibit microbial emergent properties[1,2]. As these multicellular communities develop, bacteria adapt and respond differently to local chemical environmental conditions (i.e., gradients of nutrients, oxygen, waste products and bacterial signalling compounds), resulting in subpopulations of cells with considerable physiological heterogeneity across spatial and temporal scales[3].

*Bacillus subtilis* has long been used as a model organism for genetic studies of the formation of different types of structured communities[4–7]. This gram-positive, motile, spore-forming ubiquitous bacterium is often found in the rhizosphere in close proximity to plants, as well as in extremely diverse environments[8–10]. It is used

commercially to produce proteins, fermented foods, biocontrol agents, and probiotics[11–14]. Conversely, it can potentially play a deleterious role, for example, the *B. subtilis* NDmed strain, isolated from a hospital endoscope washer-disinfector, is capable of forming biofilms with complex protruding structures that are hyperresistant to the action of oxidising agents used for endoscope disinfection, thus protecting pathogenic bacteria such as *Staphylococcus aureus* in mixed-species biofilms[15–17]. Understanding how these surface-associated communities form and interact is therefore crucial to the development of appropriate strategies to control them.

Studies of multicellular *B. subtilis* communities are typically based on the development of a floating pellicle at the air-liquid interface, a submerged biofilm at the solid–liquid interface, or the development of

[1]Université Paris-Saclay, INRAE, AgroParisTech, Micalis Institute, Jouy-en-Josas, France. [2]Lebanese University, Faculty of Science, Beirut, Lebanon. [3]Université Paris-Saclay, Centre National de la Recherche Scientifique (CNRS), INRAE, AgroParisTech, Micalis Institute, Jouy-en-Josas, France. [4]Université Paris-Saclay, INRAE, MAIAGE, Jouy-en-Josas, France. [5]Sorbonne Université, CNRS, Institut de Biologie Paris-Seine, Laboratoire Jean Perrin, Paris, France. ✉e-mail: kassem.hamze@ul.edu.lb; romain.briandet@inrae.fr

a complex colony at the solid-air interface[4,5,18]. Under certain conditions, such as on a semisolid surface, colony-forming *B. subtilis* cells can become highly motile and swarm across the surface in an organised collective movement while proliferating and consuming nutrients[19]. On a synthetic minimal medium, *B. subtilis* swarms from the mother colony in a branched, monolayer, dendritic pattern that continues to grow up to 1.5 cm from the swarm front. A transition from a monolayer swarm to a multilayer biofilm occurs from the base of the dendrite, and the biofilm spreads outwards in response to environmental cues[20–24]. The *B. subtilis* NDmed strain has been well characterised phenotypically by multiculture approaches, revealing its remarkable ability to form 3D structures, including macrocolonies, pellicles and submerged biofilms, as well as to exhibit swarming behaviour[5,7,25].

In a *B. subtilis* culture forming a biofilm, various cell types coexist[1,26–28], leading to differential regulation of several genes. This complex gene expression enables the division of labour among cells[26,29–33]. A comprehensive investigation of metabolic changes during pellicle development using metabolomic, transcriptomic and proteomic analyses revealed that metabolic remodelling is predominantly controlled at the transcriptional level[34]. Furthermore, an ontogeny study of *B. subtilis* macrocolonies grown on agar revealed a correlation with evolution, showing a temporal order of expression from older to newer genes[35]. Recently, we conducted a transcriptional study of the *B. subtilis* NDmed strain using a static liquid model in a microplate well. The study involved sampling well content over a temporal scale, providing valuable insight into the expression profiles during the first 7 h of submerged biofilm development. In addition, after 24 h of incubation, we examined mixtures of different localised populations, including submerged biofilm cells, detached cells and pellicles[36].

In the present study, our main objective was to identify the differential expression of genes specifically associated with different populations formed on solid, semisolid or liquid surfaces. To achieve this, we performed a spatial transcriptional analysis at the mesoscopic scale, focusing on seven different localised populations: four selected from a swarming model and three selected from a static liquid model. Additionally, we selected two more populations from a planktonic culture. Through this approach, we obtained a comprehensive and global landscape characterisation of gene expression for each population. This enabled us to pinpoint genes of interest, whose expression was subsequently reported using live-cell fluorescent imaging through time-lapse CLSM. Particular attention was given to understanding the single-cell scale dynamics within the submerged biofilm population.

## Results

### RNA sequencing reveals spatially resolved populations with distinct patterns of gene expression

RNA-seq was used to compare transcriptomic profiles between populations selected from the different multicellular models formed by *B. subtilis* NDmed; these populations are schematised in Fig. 1a. We considered a 24-h static liquid model from which we separately collected the submerged biofilm (SB), floating pellicle (PL) and free detached cells (DC). Furthermore, from a swarming model, four differently localised populations were collected: the mother colony (MC), the base of the dendrites (BS), the dendrites (DT) and the tips (TP); the latter three corresponded to the swarmers (Supplementary Movie 1, Supplementary Movie 2, Supplementary Movie 3). From the planktonic culture, the exponential (EX) and stationary (ST) phases were collected, the latter being used as the inoculum to initiate the two models.

A hierarchical clustering analysis of the RNA-seq data was conducted to assess data quality and reproducibility (Fig. 1b). This analysis shows how the three biological replicates were grouped, with the exception of the adjacent swarmer populations, where the clustering displayed a tendency to bring together either base and dendrites or

dendrites and tips. This discrepancy could be due to similar physiological characteristics of cells in the dendrites as those in the base or tips. To investigate these global differences, a statistical analysis was conducted to identify differentially expressed genes (DEGs) between the populations of each model (Supplementary Fig. 1). Comparing dendrites to base and tips to dendrites revealed 12 and 24 DEGs, respectively. However, the number of DEGs increased significantly to 304 when comparing the base to the tips, indicating an equivalent physiological proximity of the dendrites to the adjacent swarmer populations.

Moreover, hierarchical clustering demonstrated that both the mother colony and the stationary phase were distinct not only from each other but also from all other selected populations (Fig. 1b). A global principal component analysis (PCA) revealed that the three different biofilm populations were grouped together but were still very distant from the stationary phase, with more than 30% of the genome differentially expressed (Fig. 1c, Supplementary Fig. 2).

The RNA-seq data for the 4028 genes of *B. subtilis* NDmed revealed significant alterations in the transcriptional landscape between the different selected populations (Fig. 1d). The complete list of statistically significant associations for up- and downregulated genes in the different populations is provided in Supplementary Data 1. A website for interactive online exploration of condition-dependent quantitative expression profiles down to single-nucleotide resolution has been created using Genoscapist[37]; these profiles can be accessed at https://genoscapist.migale.inrae.fr/seb_bsubbiofilm/. Below, we highlight some gene expression changes grouped with respect to known *B. subtilis* regulons and functional categories as defined in the *Subti*Wiki database (http://subtiwiki.uni-goettingen.de)[38].

The expression levels of genes dependent on the alternative sporulation-specific sigma factors (SigE, SigF, SigG and SigK), the stress-responsive sigma factor SigB as well as most of genes from the functional category "motility and chemotaxis" controlled by sigma factor SigD showed variability in the different selected populations (clusters G1, G4, and G7 in Fig. 1d, Supplementary Fig. 3, Supplementary Data 2). Conversely, genes controlled by the housekeeping sigma factor SigA exhibited stable expression in all selected populations (cluster G2, Supplementary Fig. 3) with the condition-specific variations attributed to activity of transcription factors.

For instance, the majority of genes involved in the various aspects of the "Cell envelope biogenesis and cell division" category (215 in total) were expressed at relatively similar levels in all selected populations. However, genes involved in the biosynthesis of cell wall teichuronic acids (*tuaA-H* operon) or teichoic acids (*tagAB* operon), controlled by the transcriptional regulator PhoP, were up- or downregulated, respectively, in both the mother colony and the stationary phase populations (cluster G1 in Fig. 1d). Almost all genes belonging to the "Genetic competence" functional category displayed similar expression in all biofilm models, with upregulation in only the planktonic populations.

The genes responsible for the biosynthesis of the siderophore bacillibactin (*dhbA-ybdZ* operon, controlled by Fur) and the biosynthesis of heme (*hemA-L* operon, controlled by PerR) were both upregulated in the static liquid model populations (cluster G3 in Fig. 1d). In addition, many genes responsible for the biosynthesis of amino acids, especially histidine, arginine, methionine/S–adenosylmethionine, lysine/threonine and branched–chain amino acids, were downregulated in the submerged biofilm population of the static liquid model (cluster G2).

ABC transporter genes responsible for amino acid uptake were weakly expressed in both detached cells and submerged populations, whereas for glutamine uptake (*glnQ-P* operon), a strong upregulation was found in the mother colony (cluster G3). For the importers of compatible solutes for osmoprotection, downregulation was detected in all static liquid model populations (e.g., the *opuCA-CD* operon; G4

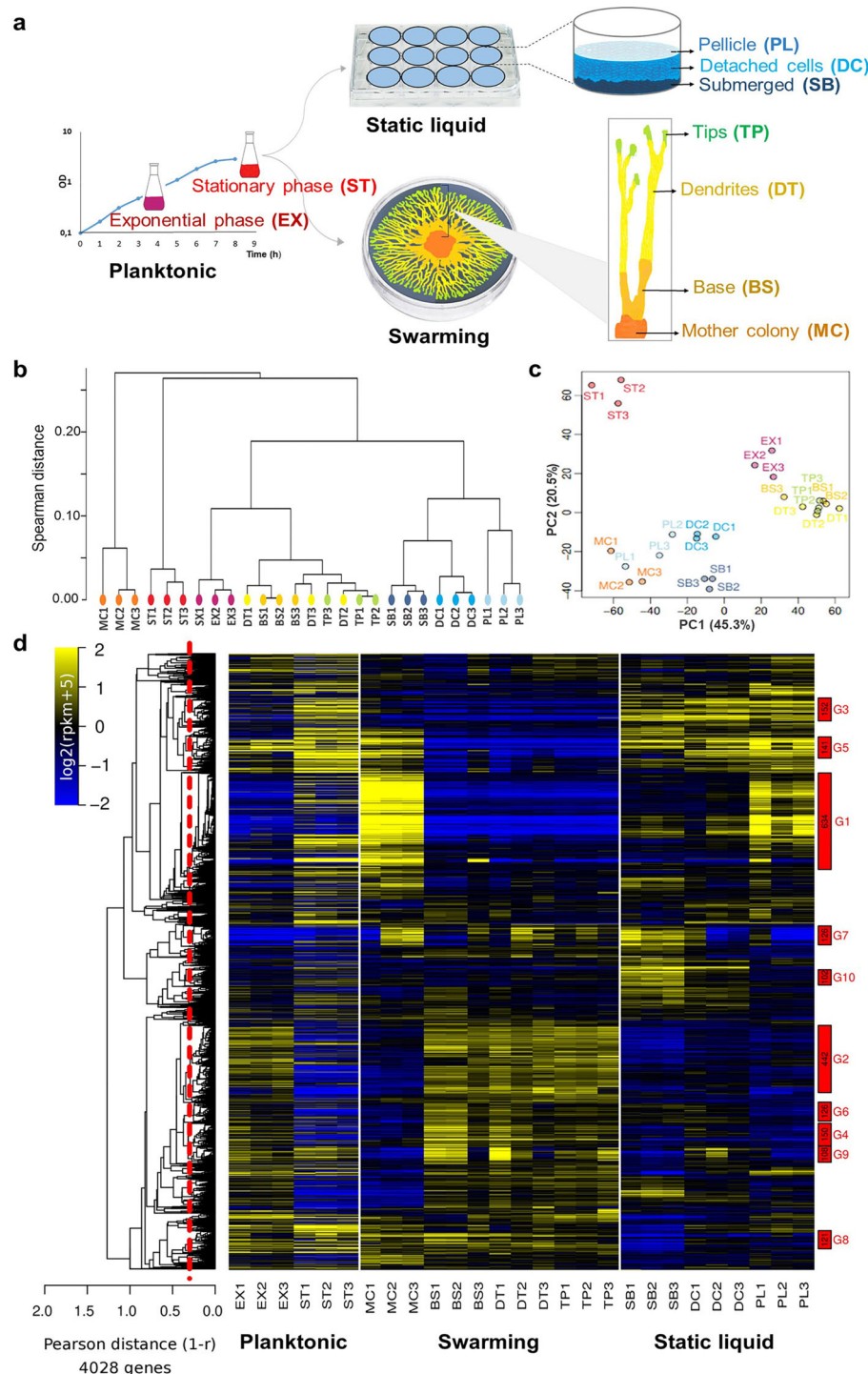

**Fig. 1 | An overview of spatial transcriptome remodelling between the different selected populations of B. subtilis. a** Schematic illustration of the differently localised spatial populations selected. From the planktonic culture, the exponential (EX) and stationary (ST) phases were selected. From the static liquid model, the pellicle (PL) formed at the liquid-air interface, the submerged biofilm (SB) formed at the solid–liquid interface, and the free detached cells (DC) between these two populations were collected separately. From the swarming model, four localised populations were collected separately: the mother colony (MC), the inoculation site from which the swarm developed as a mature macrocolony; the base (BS) of the dendrites as an earlier biofilm form; the dendrites (DT), a monolayer of cells ready to later form the biofilm; and the tips (TP) formed by motile and rapidly dividing cells. For each population, distinguished by a specific distinct colour, three independent samples were taken as biological replicates. **(b)** The pairwise distance (Spearman) between RNA-seq profiles is summarised by a hierarchical clustering tree highlighting the divergence of the mother colony (MC) and the stationary

phase (ST) between/along other selected populations and the proximity of adjacent spatial populations of either the static liquid model (SB, DC and PL) or the swarmer populations (BS, DT and TP), which share a closer genetic expression profile with the exponential phase (EX). **(c)** Projection of the 27 transcriptomes on the 2 main axes of the principal component analysis (PCA) plot. These axes account for up to 65% of the total variability of the data. **(d)** Global heatmap representation of the 4028 genes present in NDmed across the spatially selected surface-associated populations. The colour code reflects the comparison to the mean calculated for each gene (log2 ratio), taking as reference the mean of all conditions, except the planktonic ones (EX and ST). The hierarchical clustering tree shown on the left side of the heatmap (average link) was cut at an average Pearson correlation of 0.7 (dashed red line) to define the expression clusters shown as rectangles on the right side of the heatmap. Clusters were named by decreasing size (from G1 to G321), and only those containing more than 100 genes are highlighted (number of genes in black, cluster name in red).

cluster in Fig. 1d). The diversity of expression patterns of transporter-encoding genes is most likely due to the affiliation of these genes to complex metabolic and resistance pathways regulated by different transcription factors.

Respiratory chain genes, mainly activated in response to oxygen limitation by the two-component response regulator ResD, were upregulated in all static liquid model populations (e.g., the *cydA-D* operon encoding cytochrome-bd ubiquinol oxidase; cluster G3 in Fig. 1d). Furthermore, genes ensuring anaerobic (nitrate) respiration (e.g., the *narG-I* and *nasD-F* operons) were specifically upregulated in the submerged biofilm and detached cells (Supplementary Data 1).

Notably, genes involved in the biosynthesis of purines (the *purE-D* operon) and pyrimidines (the *pyrR-E* operon) were strongly down-regulated in the mother colony population compared to all other biofilm populations (cluster G11, Supplementary Data 2). These features are most likely related to the activation of the stringent response mediated by the burst of the alarmone guanosine-(penta)tetra-phosphate ((p)ppGpp) and the decrease in the nucleoside triphosphate GTP.

All biofilm populations exhibited increased expression of several genes encoding extracellular proteases (*aprE, bpr, mpr, nprB, nprE*), which ensure the degradation of cellular and environmental proteins as a source of amino acids and peptides (cluster G1).

In general, genes controlling the use of specific carbon sources were found to be expressed at similar levels in all biofilm populations, although there were some exceptions: the *lutA-C* operon (lactate utilisation) and the *csn* and *pelB* genes (chitin and polygalacturonic acid degradation, respectively) were strongly upregulated in all static liquid populations; the *xylAB* and *xynPB* operons of the XylR regulon (utilisation and degradation of xylan and xylose) were expressed at low levels in swarmer populations but upregulated in all other populations, reaching the highest expression in the pellicle (cluster G3, Fig. 1d).

The expression patterns of most genes in the "Sporulation" functional category (421 out of 629) were characterised by the highest expression levels detected in the mother colony population (cluster G1 in Fig. 1d). Cells within the latter population express genes controlled by the either early or late sporulation-specific sigma factors SigE (158 out of 221), SigF (64 out of 107), SigG (102 out of 135) and SigK (67 out of 120) as well as by relevant transcription factors (GerE, SpoVT, SpoIIID), indicating high heterogeneity with respect to the sporulation process. Interestingly, a simultaneous but less important induction of genes controlled by early (SigE and SigF) and late (SigG and SigK) sigma factors was also found in selected static liquid populations in contrast to the swarmer populations, where the expression of these genes was equivalent to that of exponentially growing cells (Supplementary Fig. 3).

Consistent with the heterogeneity of SigB-dependent expression[39], which can contribute to a population's adaptability to variations in environmental conditions, genes controlled by SigB and encoding general stress proteins were differentially and heterogeneously expressed in all biofilm populations (cluster G7, Fig. 1d). This may have contributed to the adaptability of the populations to different environmental conditions. Genes in the category "Prophage and mobile genetic elements" consistently displayed low stable transcription levels.

Most of the genes in the "Motility and chemotaxis" category, such as the *fla/che* and *motAB* operons, were strongly downregulated in all biofilm populations compared to the swarmers (cluster G9, Fig. 1d). Surprisingly, the *epsA-O* and *tapA-tasA* operons, which encode key components of the biofilm matrix involved in biofilm formation, were downregulated in all static liquid model populations, particularly in the submerged biofilm, compared to the swarming model (cluster G8, Fig. 1d).

Approximately a quarter of the genome consists of genes encoding either unknown or poorly characterised functions. Among

these, certain genes exhibited intriguing expression profiles. For instance, *yjfA*, *yezF*, and a large *yodT-yokU* operon displayed significant expression across all biofilm populations, particularly for the mother colony and pellicle (cluster G1, Fig. 1d, Supplementary Fig. 4). Conversely, *yfmQ* showed high expression in all static liquid model populations. In contrast, the *yhfTS* and *yoaD-B* operons exhibited considerable downregulation in submerged biofilms in comparison to the other populations. Notably, *yrhG* displayed an intriguing gradual upregulation pattern among the swarmer populations (Supplementary Fig. 4).

To better compare gene expression levels between adjacent populations and to highlight the different functional categories encoded by the differentially expressed genes on a spatial scale, the static liquid and swarming models were analysed individually, as shown in Supplementary Fig. 5, Supplementary Fig. 6, Supplementary Note 1, Supplementary Note 2.

## Spatial transcriptomics detected mutually exclusive carbon metabolic regimes occurring within a biofilm

Glycolysis and gluconeogenesis are two opposite pathways, for which *B. subtilis* possesses two distinct glyceraldehyde-3-phosphate dehydrogenases (GAPDH) (EC 1.2.1.12) that catalyse either the oxidative phosphorylation of glyceraldehyde-3-phosphate to 1,3-diphosphoglycerate or the reverse reaction: (i) GapA, a strictly NAD-dependent GAPDH involved in glycolysis, and (ii) GapB, which is involved in gluconeogenesis and exhibits cofactor specificity for NADP[40]. Because the coexistence of the two pathways in the same cell dissipates energy in a futile cycle[41], the expression of *gapA* and *gapB* is subjected to very efficient opposite regulations: *gapA* transcription is induced in glycolytic conditions and is repressed during gluconeogenesis by the self-regulated CggR repressor of the *cggR-gapA* operon, whereas *gapB* is transcribed only during gluconeogenesis and strongly repressed under glycolytic conditions by the CcpN repressor, as is *pckA*, which encodes the purely gluconeogenic PEP-carboxykinase (PEP-CK)[40,42,43].

Although the cultures for this study were performed under purely glycolytic conditions (*i.e.*, with glucose as a carbon source), the expression of both *gapB* and *pckA* was derepressed in the three biofilm populations after 24 h of incubation, indicating glucose depletion (Fig. 2a). Interestingly, the glycolytic *cggR-gapA* operon and the gluconeogenic *gapB*/*pckA* genes were strictly oppositely regulated in all the selected populations, except for the submerged biofilm, in which both were upregulated (Fig. 2a). This observation suggested the coexistence of two cell types in the same population, which motivated the construction of a strain reporting the expression of both *cggR-gapA* and *gapB* by different fluorescent transcriptional fusions (Table 1).

Using this strain with live-cell fluorescent imaging, we observed the in situ expression of both *cggR-gapA* and *gapB* in the different spatially localised populations at the single-cell level after 24 h (Fig. 2b). These observations confirmed the transcriptome data, with the glycolytic genes being upregulated in the swarmer populations and the gluconeogenic genes being repressed, whereas opposite expression patterns were observed in the mother colony. Similarly, in the static liquid model, the glycolytic and gluconeogenic genes were oppositely regulated in the detached cells and pellicle but were both upregulated in the submerged biofilm. Closer observation of the base in the swarming model or the pellicle in the static liquid model allowed us to visualise a few scattered cells expressing gluconeogenic genes within a larger population growing under glycolytic metabolism (Supplementary Fig. 7). We then performed in situ spatiotemporal single-scale monitoring of the submerged population at a higher resolution (Fig. 2c).

4D confocal imaging showed high expression of glycolytic genes in bacteria growing in glycolytic B medium (Supplementary Movie 4).

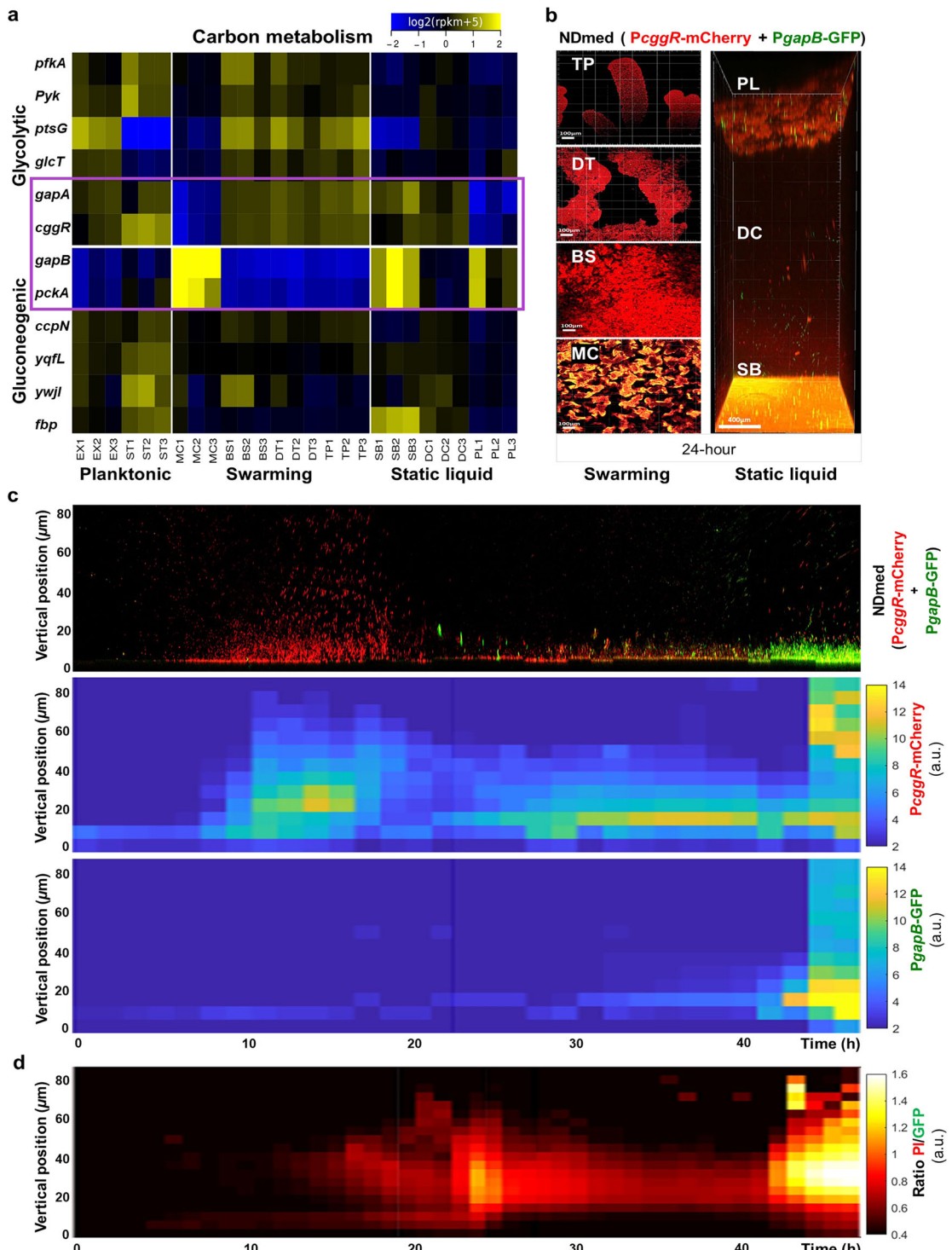

**Fig. 2 | Spatial transcriptomic remodelling with in situ imaging reveals heterogeneous differential expression of central carbon metabolism. a** Heatmap representation of the relative variation in expression levels between populations. The colour code reflects the comparison to the mean calculated for each gene across all populations, except the planktonic populations (EX and ST) (log2 ratio). Genes were selected from SubtiWiki categories specific to glycolysis or gluconeogenesis. The purple box highlights central genes specific for glycolysis (*gapA*, *cggR*) and gluconeogenesis (*gapB*, *pckA*). **(b)** Spatial confocal imaging for the different selected populations from the swarming (MC, BS, DT, TP) and static liquid (SB, DC, PL) models after 24 h at 30 °C, using strain GM3900 reporting the transcription of *cggR-gapA* by mCherry (in red) and of *gapB* by Gfp (in green), with the same protocol as for the transcriptome analysis, except for the use of a 96-well

microplate instead of a 12-well microplate for the static liquid model. Three replicate observations were performed independently for each model. **(c)** Sections from real-time confocal imaging (x 50 μm, y 50 μm, z 80 μm) for 48 h, imaged every one and a half h, using strain GM3900 reporting the transcription of *cggR-gapA*) by mCherry (in red) and of *gapB* by Gfp (in green) (Supplementary Movie 4). Kymographs representing the intensity of the expression of the transcriptional reporter fusions to the *cggR* and *gapB* promoters along a spatiotemporal scale by colour coding. Three biological replicates were performed (Supplementary Fig. 13). **(d)** A multidimensional kymograph representing the intensity of dead cells, obtained from the ratio of dead/living cells, as a function of their spatial localisation and time (Supplementary Movie 5). The kymograph is representative of at least three replicates (Supplementary Fig. 12).

**Table 1 | *B. subtilis* strains used in this study**

| Strain | Relevant genotype or isolation source | Construction or Reference[a] |
|---|---|---|
| NDmed | Non-Domesticated wild type strain | 15 |
| NDmed-GFP GM3649 | NDmed *amyE::Phyperspank-gfpmut2* (spec) | 16 |
| BSB168 | *trp+* derivative of 168 | 63,64 |
| BBA093 | BSB168 *PackA-gfpmut3* (spec) | Matthieu Jules[b] |
| BBA0231 | BSB168 *Phag-gfpmut3* (spec) | Matthieu Jules[b] |
| BBA0290 | BSB168 *PbslA-gfpmut3* (spec) | Matthieu Jules[b] |
| BBA0428 | BSB168 *PsrfAA-gfpmut3* (spec) | Matthieu Jules[b] |
| BBA9006 | BSB168 *PgapB-gfpmut3* (spec) | Matthieu Jules[b][65] |
| GM3346 | NDmed *Phag-gfpmut3* (spec) | BBA0231→NDmed |
| GM3348 | NDmed *PackA-gfpmut3* (spec) | BBA0093→NDmed |
| GM3378 | NDmed *PgapB-gfpmut3* (spec) | BBA9006→NDmed |
| GM3401 | NDmed *PbslA-gfpmut3* (spec) | BBA0290→NDmed |
| GM3402 | BSB168 *PepsA-gfpmut3* (spec) | pBSB2epsA → BSB168 |
| GM3403 | NDmed *PsrfAA-gfpmut3* (spec) | BBA0428 →NDmed |
| GM3423 | NDmed *PepsA-gfpmut3* (spec) | GM3402→NDmed |
| GM3461 | BSB168 *PypqP-gfpmut3* (spec) | pBSB2ypqP → BSB168 |
| GM3476 | NDmed *PypqP-gfpmut3* (spec) | GM3461→NDmed |
| GM3816 | NDmed *PctaA-gfpmut3* (spec) | pBSB2ctaA→NDmed |
| GM3820 | NDmed *PnarG-mCherry* (cm) | pBSB8narG→NDmed |
| GM3823 | NDmed *PskfA-mCherry* (cm) | pBSB8skfA→NDmed |
| GM3838 | NDmed *PcomGA-gfpmut3* (spec) | pBSB2comGA→NDmed |
| GM3841 | NDmed *PaprE-mCherry* (cm) | pBSB8aprE→NDmed |
| GM3859 | NDmed *PcggR-mCherry* (cm) | pBSB8cggR→NDmed |
| GM3862 | NDmed *capE-mCherry* (cm) | pBSB8capE→NDmed |
| GM3864 | NDmed *PspoIIGA-mCherry* (cm) | pBSB8spoIIGA→NDmed |
| GM3867 | NDmed *PspoVC-mCherry* (cm) | pBSB8spoVC→NDmed |
| GM3874 | NDmed *PtapA-mCherry* (cm) | pBSB8tapA→NDmed |
| GM3900 | NDmed *PgapB-gfpmut3* (spec)/ *PcggR-mCherry* (cm) | GM3859 → GM3378 |
| GM3903 | NDmed *PaprE-mCherry* (cm)/ *PackA-gfpmut3* (spec) | GM3841 → GM3348 |
| GM3907 | NDmed *PnarG-mCherry* (cm)/ *PctaA-gfpmut3* (spec) | GM3820 → GM3816 |
| GM3912 | NDmed *PskfA-mCherry* (cm)/ *PcomGA-gfpmut3* (spec) | GM3838 → GM3823 |
| GM3924 | NDmed *PtapA-mCherry* (cm)/ *Phag-gfpmut3* (spec) | GM3346 → GM3874 |

[a]arrows indicate transformation of pointed strain with indicated plasmid or chromosomal DNA of indicated strain.
[b]Université Paris-Saclay, INRAE, AgroParisTech, Micalis Institute, 78350 Jouy-en-Josas, France.

Upon glucose consumption, a gradual decrease in their expression is followed by a sudden derepression of gluconeogenic genes in small clusters of few cells. After 24 h, the cells regained glycolytic metabolism with the persistence of some clusters of cells under gluconeogenesis (Fig. 2c) and with a subsequent strong increase in both subpopulations expressing opposite carbon metabolism pathways (Supplementary Movie 4). This led us to question whether this regaining of glycolytic metabolism could be due to dead cells being a source of some glycolytic metabolites.

To test this hypothesis, microscopic fluorescent live/dead tracking was performed at the single-cell level. Bacteria adhered to the surface and formed chains of sessile cells during the first hours of incubation, and then, between approximately 15 and 24 h, clusters of dead cells were observed above the formed biofilm (Fig. 2d, Supplementary Movie 5). After this first wave, the density of dead cells decreased, with a slight increase in the live population until approximately 42 h, when a second wave of dead cells was observed

(Fig. 2d, Supplementary Movie 5). By aligning the metabolic and dead cell kymographs, it is clear that the regain of glycolytic metabolism directly followed the peak of cell death (Fig. 2c and d). A similar phenomenon was observed when glucose was replaced by glycerol, another glycolytic carbon source (Supplementary Fig. 8). To test the effect of a purely gluconeogenic carbon source, glucose was also replaced by malate. In this condition, cell growth under the gluconeogenic regime was followed by the emergence of a subpopulation of cells under the glycolytic regime, which also seems to be correlated with the occurrence of cell death (Supplementary Fig. 8). Thus, the coexistence of spatially mixed subpopulations under opposite metabolic regimes reveals metabolic exchanges within the submerged biofilm.

**Live-cell fluorescence imaging illuminates spatiotemporal mosaic patterns of gene expression within submerged biofilms**

Based on the transcriptome data, the submerged biofilm exhibited downregulation of major matrix genes compared to the other biofilm populations (Supplementary Data 1, Fig. 1d). This suggests that these genes were either consistently expressed at a low level within the submerged biofilm or were expressed only during a brief period before or after the 24-h transcriptome time point. To further investigate the characteristics of the submerged biofilm, we opted to observe these particular genes and other genes involved in various biological functions that could potentially influence biofilm development; these chosen genes were distributed among different clusters within the global heatmap (Fig. 1d, Supplementary Fig. 9). Using fluorescent reporter transcriptional fusions, we closely monitored the expression of these genes/operons involved in matrix synthesis (*epsA-O, tapA-tasA, bslA, srfAA-AD, ypqP, capA-E*), motility (*hag*), exoprotease synthesis (*aprE*), carbon metabolism (*ackA*), competence (*comGA-GG*), cannibalism (*skfA*), respiration (*ctaA, narG-I*), and sporulation (*spoIIGA-sigG, spoVC-VT*).

In Fig. 3a, a fluorescence-based kymograph illustrates the kinetics of submerged biofilm formation by the NDmed-GFP strain, which serves as a reference for the subsequent gene expression observations presented in Fig. 3b. Each reported gene showed different expression patterns during submerged biofilm formation, reflecting the specificity of their functions. The expression of the *epsA-O* and *tapA-tasA* operons peaked at an early stage, within the first hours of incubation, initiating matrix development. As the submerged biofilm continued to grow, the basal uniform level of *bslA* expression gradually increased, while the *srfAA-AD* operon showed strong expression at an intermediate stage of development. In contrast, *ypqP* and the *capB-E* operon were expressed at a late stage of submerged biofilm formation. Comparably, the expression of *hag*, involved in motility, was upregulated synchronously with the beginning of the downregulation of *tapA-tasA* and the gradual upregulation of *narG-I* (Supplementary Movie 6, Supplementary Fig. 10, Supplementary Fig. 11). Oscillating expression of *hag* was also observed during the 48 h (Supplementary Movie 7). The other reported genes/operons showed either gradual downregulation (*ackA*) or gradual upregulation (*ctaA, aprE, comGA-GG, skfA, spoIIGA-sigG* and *spoVC-VT*).

Similar to the *cggR-gapA* operon (reporting glycolysis), *ackA* was highly expressed during the first 13 h before being gradually downregulated, indicating the onset of glucose limitation (Fig. 2c and Fig. 3, Supplementary Movie 8). This could trigger the expression of *spoIIGA-sigG*, indicating entry into the irreversible process of sporulation, as well as the gradual expression of *skfA*, leading to the production of the spore-killing factor (Supplementary Movie 9). As a consequence, a first wave of cell death was observed (Fig. 2d), presumably releasing exogenous nutrients that could be used for cell growth. Interestingly, this wave was not observed with a *skfA* mutant (Supplementary Fig. 12), suggesting a temporal correlation between cell death and SkfA production.

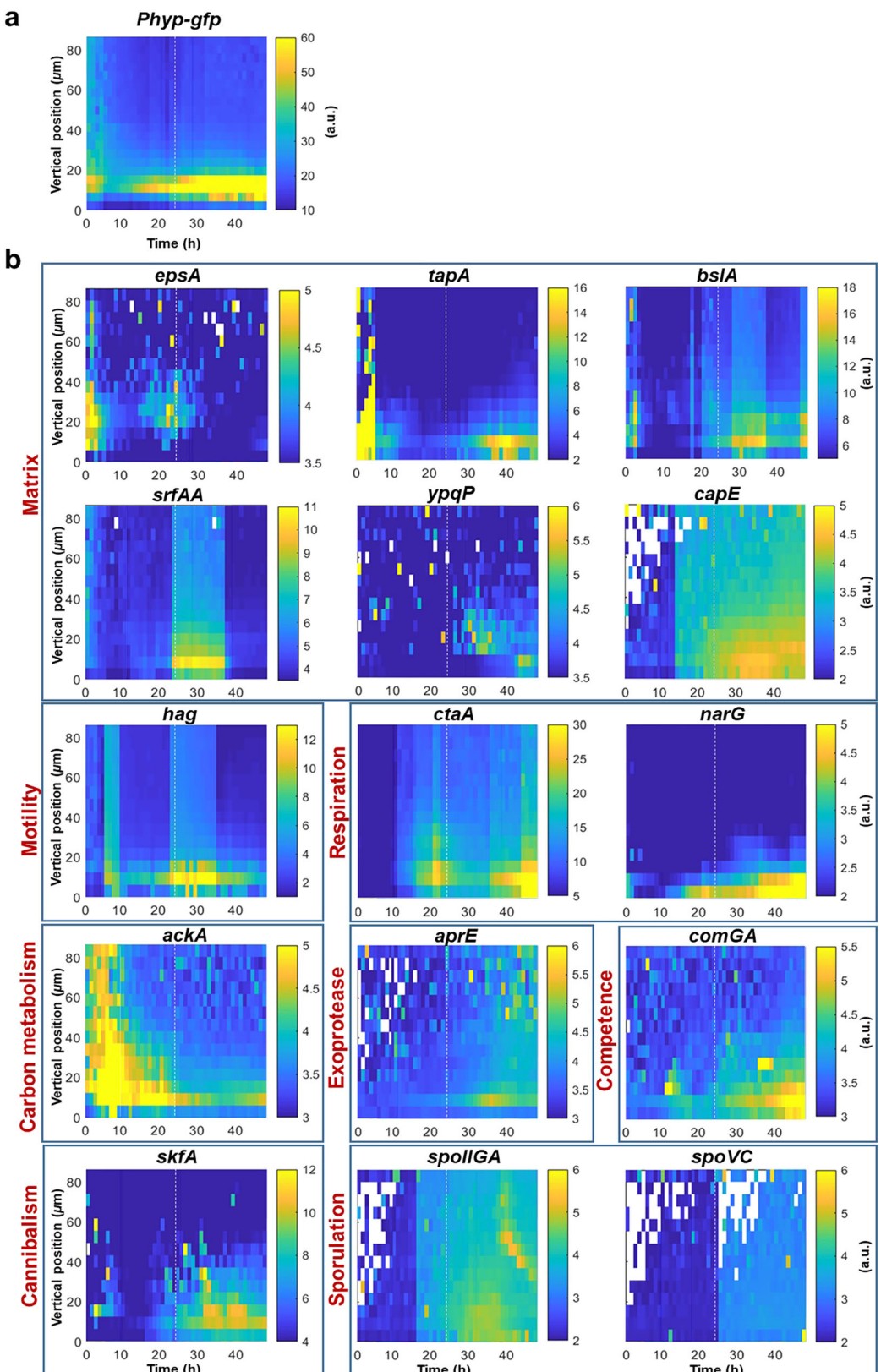

**Fig. 3 | Space-time kymographs of fluorescent reporters for the transcription of 15 genes in the submerged biofilm (SB). a** Kymograph of the fluorescent signal in arbitrary units (a.u.) of the constitutively expressed Gfpin the NDmed-GFP strain. **b** Kymographs for the 15 transcriptionally reported genes/operons, representing the intensity of their expression. The white dotted line in each kymograph represents the time (24 h) corresponding to the RNA-seq analysis. The white spots in the graphs indicate intensity values below a threshold limit, representing no significant signal for that specific tile. Each kymograph presented is one of three independent biological replicates shown in Supplementary Fig. 13.

Figure 3b displays the gradual expression of *aprE* and *comGA-GG* that occurred after the wave of cell death, indicating the ability of the cells to degrade and take up the released exogenous nutrients (Supplementary Movie 8 and Supplementary Movie 9). After 24 h, cells undergoing sporulation completed their process, as indicated by the overexpression of the late sporulation gene *spoVC-VT*. These observations revealed that gene expression in submerged biofilms is not uniform but rather exhibits dynamic spatiotemporal mosaic patterns.

## Discussion

This report presents a comparative description of complete transcriptomic profiles across nine spatio-physiological populations of the *B. subtilis* NDmed strain captured on solid, semisolid, and liquid cultures. This study allowed us to specify the singularities of each population and accurately determine the intricacies of their spatiotemporal regulation down to the single-cell scale. A uniqueness of this work lies in the consistent utilisation of a defined medium across all the different cultures. This medium not only allowed NDmed to exhibit swarming behaviour with dendritic patterns originating from a colony[7], but also enabled the development of other biofilm types (pellicle and submerged) in a static liquid model. Furthermore, our investigation extends to the widely studied NCIB3610 strain, which is commonly employed in biofilm research[1,34,35]. Encouragingly, this strain exhibited analogous behaviours within the defined medium used under all distinct culture conditions (Supplementary Movie 10, Supplementary Fig. 14).

Despite the diverse environmental conditions, certain similarities emerged within distinct biofilm populations under scrutiny. Notably, the *hag* gene, encoding flagellin and reporting motility, displayed reduced activity across all three biofilm types (mother colony, submerged and pellicle) compared to the other populations studied. On the other hand, genes associated with the matrix were more strongly and homogeneously expressed in the aerial biofilms than in the submerged biofilm. The diversity in the spatial repartition of cells producing these matrix components suggests the existence of distinct biochemical matrix compositions associated with specific local microrheological attributes. Leveraging live-imaging techniques alongside transcriptional fluorescent fusions that track the expression of both *hag* and *tapA-tasA* within individual cells allowed us to reveal the spatialisation of minor cells expressing the motility genes tucked beneath matrix-producing cells within biofilms (Supplementary Fig. 15). Within the context of the swarming model, swarmers give rise to spatiotemporal populations. Analysis of transcriptomic data revealed pronounced expression of motility and chemotaxis genes, underscoring the exploratory nature of the cells as they navigate towards nutrient-rich regions (Supplementary Movie 1, Supplementary Movie 2, Supplementary Movie 3); in stark contrast, cells in the mother colony exhibited elevated expression levels of matrix genes. In the static liquid model, a two-phase process has been previously observed when tracking the dynamics of motile/sessile gene expression: initially, cells adhere to the submerged surface and form elongated chains, followed by sudden fragmentation to motile cells, ultimately resulting in the formation of a pellicle at the air-liquid interface[36]. This fragmentation coincides with oxygen depletion, followed by reorganisation of the submerged biofilm under anaerobic metabolism[36]. Strikingly, this behaviour, initially described in a rich medium, mirrors a similar phenomenon within the defined medium employed in our study (Supplementary Movie 6, Supplementary Fig. 10, Supplementary Fig. 11). This observation underscores the coexistence of two interfacial biofilm communities of *B. subtilis*, each characterised by distinct respiratory metabolisms: the submerged biofilm, along with detached cells, predominantly operated under anaerobic conditions, while the pellicle thrived through aerobic respiration. As a result, while these biofilm populations shared certain parallels, they diverged in key aspects, such as the prevalence of anaerobic respiration metabolism within the submerged biofilm, contrasting with the mother colony and

pellicle. Remarkably, despite the direct contact with air of the latter biofilm populations, isolated cells expressing anaerobic genes still persisted (Supplementary Fig. 15).

Moreover, the genes involved in sporulation exhibited robust upregulation within the aerobic biofilms, while their expression remains suppressed within the submerged population. Intriguingly, a specific spore count performed within the static liquid model revealed a higher number of spores within the submerged biofilm fraction than in the floating pellicle (Supplementary Fig. 16). This apparent inconsistency may be explained by the sedimentation phenomenon, where spores produced within the pellicle tend to settle due to their hydrophilic nature, accumulating within the submerged layer. This behaviour could be attributed to the presence of legionaminic acid coating the *B. subtilis* spores, a requisite for crust assembly and heightened cell wall hydrophilicity[44]. Live-cell imaging of the expression of sporulation genes within the submerged biofilm revealed the activation of late sporulation genes after 24 h of incubation (Fig. 3b). Preceding this, the onset of early sporulation gene expression coincided with the depletion of glycolytic carbon source (as indicated by the differential expression of the oppositely regulated *cggR-gapA* and *gapB* genes) and a shift towards gluconeogenic metabolism. Subsequently, a gradual production of SkfA coincided with the emergence of an initial wave of cell death atop the viable cells (Supplementary Fig. 13). Notably, this wave of cell death remained absent within the context of a *skfA* mutant at that given time, in contrast to the second unaffected wave. Interestingly, this mutant context led to a distinctive growth trajectory within planktonic culture in the same medium, marked by a plateau during the stationary phase, culminating in cell lysis. This diverges from the behaviour of the wild-type counterpart, which resumed growth (Supplementary Fig. 17). Additionally, the *skfA* mutant colony lacked the characteristic wrinkles seen in the wild-type NDmed in response to mechanical forces resulting from increased cell density (Supplementary Fig. 18). Notably, in the wild-type colony, dead cells accumulated beneath these wrinkles, positioned at the base of the biofilm near the agar. This accumulation contributed to the creation of channels, thereby augmenting liquid transport within the colony[45–47].

The detached cells, a population between the two interfacial biofilms in the static liquid model, have long been likened to planktonic cells. Insight from phenotypic and transcriptomic investigations across diverse bacterial species, including *Klebsiella pneumoniae* and *Streptococcus pneumoniae*, has illuminated the distinct gene expression profile of detached cells, setting them apart from both sessile and planktonic cells[48–51]. Our findings with *B. subtilis* corroborate these previous observations, as the transcriptomic landscape of detached cells revealed a distinctive state that diverges from that of planktonic cells. Importantly, this state aligns more closely with the physiological characteristics of cells experiencing exponential growth than those in the stationary phase (Figs. 1b, c and Supplementary Fig. 1).

This study significantly enhances our comprehension of the diversity in cellular gene expression and functional behaviours observed among the distinct spatial biofilm populations. However, the intricate spatial transcriptional patterns derived from these diverse populations provide only a glimpse into the potential cellular variations that unfold during biofilm development. Nonetheless, by coupling spatial transcriptomic analysis with live-cell imaging using confocal microscopy, we have gleaned invaluable insights regarding both the spatial arrangement and temporal changes within *B. subtilis* biofilms, with a special emphasis on submerged biofilms. Spatially resolved transcriptomics was acknowledged as the Nature method of the year 2020[52], primarily for its remarkable advancements in the realm of eukaryotic cell and tissue research. Despite this recognition, research into spatially organised microbial communities remains relatively underexplored. Furthermore, genome-scale transcriptomes can be accessed even within localised bacterial populations comprising just a few thousand individuals. The precision of population extraction can be heightened using laser

capture microdissection microscopy (LCMM)[53]. Innovative technologies such as parallel sequential fluorescence in situ hybridisation (par-seq-FISH) enable the mapping of gene expression in single biofilm layers[54]. The incorporation of live-cell fluorescent imaging in this study revealed spatiotemporal gene expression patterns, requiring extensive genetic manipulations. Integrating these multifaceted approaches offers a gateway to delve into microscale single-cell phenotypic disparities in biofilms, illuminating complex interactions in heterogeneous and dynamic environments.

Beyond the wealth of knowledge gained from this research lies the potential to devise strategies for controlling or harnessing biofilm formation and dispersal in diverse fields such as biotechnology, environmental science, and medicine. Additionally, the comprehensive dataset generated in this study serves as a valuable and unique resource for future investigations into the genetic determinants and regulatory networks governing biofilm behaviour, even unravelling the functions of genes heretofore unknown.

## Methods
### Bacterial strains and growth conditions
The *B. subtilis* strains utilised in this study are listed in Table 1. NDmed derivatives were obtained by transforming NDmed with various plasmids or chromosomal DNA from different strains to introduce the corresponding suitable reporter fusion. The transcriptional fusions of the *gfpmut3* gene to the *ackA*, *hag*, *bslA*, *srfAA* or *gapB* promoter were previously constructed within the pBSB2 plasmid (pBaSysBioII) using ligation-independent cloning[55]. Subsequently, they were integrated into the chromosome of BSB168 in a nonmutagenic manner, resulting in the generation of strains BBA0093, BBA0231, BBA0290, BBA0428 and BBA9006. The chromosomal DNA of each strain was used to transfer the corresponding fusion into NDmed by transformation. Similarly, fragments corresponding to the promoter regions of *epsA*, *ypqP*, *ctaA*, *narG*, *skfA*, *comGA*, *aprE*, *cggR*, *spoIIGA*, *spoVC*, and *tapA* or to a region in the 3' part of *capE* for the reporting of the *capB-E* operon were amplified by PCR from genomic DNA using appropriate pairs of primers (Supplementary Table 1). These fragments were inserted by ligation-independent cloning in pBSB2 or in pBSB8, a pBSB2 derivative with the *gfpmut3* and *spec* (spectinomycin resistance) genes replaced by *mCherry* (codon-optimised for *B. subtilis*) and *cm* (chloramphenicol resistance), respectively. The resulting plasmids were then used to integrate each corresponding transcriptional fusion into the chromosome of *B. subtilis* through single recombination. Transformation of *B. subtilis* was performed following standard procedures, and the transformants were selected on Luria–Bertani (LB, Sigma, France) plates supplemented with appropriate antibiotics at the following concentrations: spectinomycin, 100 µg/mL; chloramphenicol, 5 µg/mL. Before each experiment, cells were cultured on Tryptone Soya Agar (TSA, BioMérieux, France). Bacteria were then grown in synthetic B-medium composed of (all final concentrations) 15 mM $(NH_4)_2SO_4$, 8 mM $MgSO_4.7H_2O$, 27 mM KCl, 7 mM sodium citrate.$2H_2O$, 50 mM Tris/HCl (pH 7.5), 2 mM $CaCl_2.2H_2O$, 1 µM $FeSO_4.7H_2O$, 10 µM $MnSO_4.4H_2O$, 0.6 mM $KH_2PO_4$, 4.5 mM glutamic acid (pH 8), 862 µM lysine, 784 µM tryptophan, 1 mM threonine and 0.5% glucose added before use[56]. Cultures for the planktonic inoculum were prepared in 10 mL of B-medium inoculated with a single colony and shaken overnight at 37 °C. The culture was then diluted to an $OD_{600nm}$ of approximately 0.1 and grown at 37 °C until it reached an $OD_{600nm}$ of approximately 0.2. The procedure was repeated twice, and finally, the culture was grown to reach the stationary phase, which was then used to inoculate cultures for swarming and static liquid models (Fig. 1a).

### Swarming culture conditions
The $OD_{600nm}$ was measured, the culture was diluted, and 2 µL of diluted bacterial culture (adjusted to an $OD_{600nm}$ of 0.01, ~$10^4$ CFU) was inoculated at the centre of a B-medium agar plate and incubated for 24 h at 30 °C with 50% relative humidity. Plates (10 cm diameter, Greiner bio-one, Austria) containing 25 mL of agar medium (0.7% agar) were prepared 1 hour before inoculation and dried with lids open for 5 minutes before inoculation.

### Liquid biofilm culture conditions
Cultures were performed in microplates, either 3 mL per well in a 12-well microplate (Greiner bio-one, Germany) or 150 µL per well in a 96-well microscopy-grade microplate (µclear, Greiner bio-one, Germany), inoculated from a stationary phase culture and adjusted to an $OD_{600nm}$ of 0.01. The plates were incubated at 30 °C for 24 h, followed by either local cell harvesting or microscopic imaging. The 96-well plate was used for kinetic monitoring of the submerged biofilm, and the pellicle was collected from a 12-well plate for observations. When necessary, the medium was supplemented with 200 µM isopropyl-β-d-thiogalactopyranoside (IPTG) to induce Gfp expression from the *Phyperspank* promoter.

### Local mesoscopic cell harvesting for RNA-seq
For the EX and ST phases ($OD_{600nm}$ ~ 0.6 and ~2.8, respectively), 6 mL of each culture was collected and pelleted by centrifugation at 8,000 × g at 4 °C for 30 seconds. The pellet was then homogenised with 500 µl of TRIzol reagent (Invitrogen, Carlsbad, CA, USA) to stabilise the RNA in the cell. For the swarming model, using 24 hour plates, four spatially localised populations (MC, BS, DT and TP) were collected independently. Collection was performed manually by using a scraper (SARSTEDT, USA) starting from the tips down to reach the mother colony (which was collected by a loop). Cells of each localised population were collected from 16 plates in an Eppendorf tube (CLEARline microtubes, Italy) containing 500 µL of TRIzol reagent. For a 24 hour static liquid model, 6 wells (from a 12-well microplate) were used to collect each sample. By using a scraper, PLs were collected in 6 mL of water. For DCs, 1 ml of the supernatant was collected from 6 wells. For SB collection, after discarding all the remaining liquid, 1 ml of water was added to a well, and cells were collected by scratching with a pipet tip. Samples were centrifuged rapidly for 30 seconds (8000 × g at 4 °C), and pellets were resuspended in 500 µL of TRIzol.

A centrifugation step for all the above collections for 1 minute to discard the TRIzol reagent was performed, and samples were snap-frozen in liquid nitrogen to be transferred to −80 °C for the RNA extraction step. For each of the 9 samples, 3 biological replicates were performed.

### RNA extraction for RNA-seq
For all nine different conditions, a washing step for the pellets of *B. subtilis* NDmed was performed with 1 mL of TE (10 mM Tris, 1 mM EDTA, pH=8) + 60 µl 1 M EDTA followed by centrifugation for 30 seconds (8000 × g at 4 °C). Cell pellets were suspended in 1 mL of TRIzol reagent. The cell suspension was transferred to a Fastprep screw cap tube containing 0.4 g of glass beads (0.1 mm). Cells were disrupted by bead beating for 40 seconds at 6.5 m/s in a FastPrep-24 instrument (MP Biomedicals, United States). The supernatant was transferred to an Eppendorf tube, and chloroform (Sigma–Aldrich, France) was added at a ratio of 1:5, followed by centrifugation at 8000 × g for 15 minutes at 4 °C. The chloroform step was repeated twice. The aqueous phase was transferred to a new Eppendorf tube, where sodium acetate (pH=5.8) was added at a final concentration of 0.3 M along with 500 µl of isopropanol (Sigma–Aldrich, France). Samples were left overnight at −20 °C and then centrifuged for 20 minutes. Pellets were washed twice with 75% ethanol (VWR, France) followed by centrifugation for 15 minutes at 4 °C. Then, pellets were dried for 5 minutes under the hood. An RNA cleanup kit (Monarch RNA Cleanup Kit T2050, New England Biolabs, France) was used to further clean the RNA samples. Extracted RNA samples were stored in RNAse/DNAse-free water (Ambion, United Kingdom) at −80 °C. Nanodrop and Bioanalyzer

instruments were used for quantity and quality controls. Library preparation, including ribosomal RNA depletion and sequencing, was performed by the I2BC platform (Gif-sur-Yvette, France) using TruSeq Total RNA Stranded and Ribo-Zero Bacteria Illumina kits, an Illumina NextSeq 550 system and NextSeq 500/550 High Output Kit v2 to generate stranded single-end reads (1 × 75 bp).

### RNA-seq data analysis

Primary data processing was performed by the I2BC platform and consisted of demultiplexing (with bcl2fastq2-2.18.12), adapter trimming (Cutadapt 1.15), quality control (FastQC v0.11.5), and mapping (BWA v0.6.2-r126)[57] against the NDmed genome sequence (NCBI WGS project accession JPVW01000000)[58]. This generated between 13 M and 29 M uniquely mapped reads per sample, which were summarised as read counts for 4028 genes (featureCounts)[59], after discarding 7 loci whose sequences also matched External RNA Controls Consortium (ERCC) references. The downstream analysis was performed using the R programming language. Samples were compared by computing pairwise Spearman correlation coefficients ($\rho$) and distance ($1$-$\rho$) on raw read counts, which were summarised by a hierarchical clustering tree (average-link). Detection of DEGs used the R package "DESeq2" (v1.30.1)[60] to estimate p- values and log2-fold-changes. To control the false discovery rate, for each pair of conditions compared, the vector of p values served to estimate q-values with the R package "fdrtool" (v1.2.16)[61]. DEGs reported for pairwise comparisons of *B. subtilis* spatial populations were based on a q-value ≤ 0.05 and, unless stated otherwise, |log2FC| ≥1. Read counts normalised per kilobase of feature length per million reads mapped (rpkm) values computed by DESeq2 based on robust estimation of library size were used to represent the expression levels for each gene in each sample. Genes were compared for their expression profiles across samples for selected sets of conditions based on pairwise Pearson correlation coefficients (r) and distance (1-r) computed on log2(rpkm+5) and average-link hierarchical clustering of the distance matrix. Accordingly, the associated heatmaps represent gene-centred variations in log2(fpkm+5) values across samples. Gene clusters defined by cutting the hierarchical clustering trees at a height of 0.3 (corresponding to average within-group r of 0.7) were numbered by decreasing number of genes coupled in the same group, with G1 being the largest. The resulting gene clusters were systematically compared to *Subti*Wiki functional categories and regulons[38] (from hierarchical level 1 to level 5) using Fisher's exact test applied to 2 × 2 matrices. The results of the comparisons with *Subti*Wiki functional categories were summarised in the form of stacked bar plots after manually assigning each gene to the most relevant category in the context of this study (when the same gene belonged to several categories) and a grouping of categories corresponding to hierarchical level 2 except for "Metabolism" (level 1) and "motility and chemotaxis" and "biofilm formation" (level 3). The whole transcriptomic dataset has been deposited in GEO (accession number GSE214964).

### Graphical representations of gene expression

Graphical representations of the expression level of a gene in a given condition were prepared by using the geometric mean of log2(rpkm +5) values. To allow interactive exploration of the sense and antisense signals along the genome with bp resolution, we used Genoscapist[37]. The bp-level signal, displayed as log2(rpkm+5), can be accessed via the website https://genoscapist.migale.inrae.fr/seb_bsubbiofilm/.

### CLSM

The biofilm models were observed using a Leica SP8 AOBS inverted confocal laser scanning microscope (CLSM, LEICA Microsystems, Wetzlar, Germany) on the INRAE MIMA2 platform (https://doi.org/10.15454/1.5572348210007727E12). For observation, strains were tagged fluorescently in green with SYTO 9 (0.5:1000 dilution in water from a

stock solution at 5 µM in DMSO; Invitrogen, France) or SYTO 61 (1:1000 dilution in water from a stock solution at 5 µM in DMSO; Invitrogen, France), two nucleic acid markers. After 15 minutes of incubation in the dark at 30 °C to enable fluorescent labelling of the bacteria, plates were then mounted on the motorised stage of the confocal microscope. For the carbon metabolism reporting genes, 3D (xyz) acquisitions were performed by an HC PL FLUOTAR 10x/0.3 DRY objective (512 × 512 pixels, pixel size 0.361 µm, 1 image every z = 20 µm with a scan speed of 600 Hz) to capture the submerged biofilm and pellicle in the same well. Moreover, the different selected populations were scanned using either HC PL APO CS2 63x/1.2 water immersion or 10x objective lenses. SYTO 9, Gfp, and IP excitation was performed at 488 nm with an argon laser, and the emitted fluorescence was recorded within the ranges of 500–550 nm and 600–750 nm on hybrid detectors. SYTO 61 or mCherry excitation was performed at 561 nm with an argon laser, and the emitted fluorescence was recorded within the range 600–750 nm on hybrid detectors. The 3D (xyz) acquisitions comprised scans at a resolution of 512 × 512 pixels, with a pixel size of 0.361 µm and an image capture every z = 1 µm interval utilising a scan speed of 600 Hz. For 4D (xyzt) acquisitions, an image was taken every 1 hour for 48 h or every 1 and a half hours for 72 h kinetics. The intensity of a calibrated reflected laser remained consistent within an acceptable range (not more than 10% variation) throughout the duration of the experiment.

The whole 4D-CLM dataset has been deposited in Data INRAE (accession number Z511A6 [https://doi.org/10.57745/Z511A6]).

### Image analysis

Projections of the biofilm, 3D or 4D, were constructed from Z-series images using IMARIS 9.3 (Bitplane, Switzerland). Space-time kymographs were constructed with the BiofilmQ visualisation toolbox from the 4D-CLSM series[62].

### Reporting summary

Further information on research design is available in the Nature Portfolio Reporting Summary linked to this article.

## Data availability

The whole transcriptomic dataset generated in this study has been deposited in GEO (accession number GSE214964). The detailed expression profiles can be visualised at https://genoscapist.migale.inrae.fr/seb_bsubbiofilm/. The whole 4D-CLM dataset has been deposited in Data INRAE (accession number Z511A6). Source data are provided with this paper.

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

## Acknowledgements
This work was supported by INRAE. Yasmine Dergham is the recipient of funding from the Union of Southern Suburbs Municipalities of Beirut, INRAE, Campus France PHC CEDRE 42280PF and Fondation AgroParisTech. Pilar Sanchez-Vizuete was the recipient of a PhD grant from the Région Ile-de-France (DIM ASTREA). Matthieu Jules (Micalis Institute) is acknowledged for the gift of plasmids pBSB2 and pBSB8 and of strains BBA0093, BBA0231, BBA0290, BBA0428 and BBA9006. Magali Calabre (Micalis Institute) is acknowledged for technical assistance. We acknowledge the sequencing and bioinformatics expertise of the I2BC High-throughput sequencing facility, supported by France Génomique (funded by the French National Program "Investissement d'Avenir" ANR-10-INBS-09). Biofilm imaging was realised at the INRAE MIMA2 imaging platform https://doi.org/10.15454/1.5572348210007727E12. We are grateful to the INRAE MIGALE bioinformatics facility (https://doi.org/10.15454/1.5572390655343293E12) for hosting the website https://genoscapist.migale.inrae.fr/seb_bsubbiofilm/. The authors extend their appreciation to Marie-Françoise Noirot-Gros, Arnaud Bridier and Manish Kushwaha for their valuable contributions and enriching discussions. The article was edited by a native English speaking editor at Springer Nature Authors Services (SNAS C131-F89A-F251-02AC-C33P). This work was performed under the umbrella of the European Space Agency Topical Team: Biofilms from an interdisciplinary perspective.

## Author contributions
Y.D., D.L.C, K.H. and R.B. designed research; Y.D., D.L.C, E.H., J.D., M.D. and P.S.V performed research; Y.D., P.N., S.D., E.B., D.L.C and R.B analysed data; Y.D., D.L.C, P.N., E.B., K.H., and R.B. wrote the manuscript with support from all authors.

## Competing interests
The authors declare no competing interests.
