## [Peer Review File · Nature Communications]

Direct comparison of spatial transcriptional heterogeneity across diverse *Bacillus subtilis* biofilm communitiesReviewer #1 (Remarks to the Author):

This manuscript presents two main datasets: First, a transcriptome dataset from *Bacillus subtilis* in different conditions, including liquid culture, swarms and different types of biofilms. Second, a spatial-temporal microscopy dataset for 17 fluorescent reporters.

The authors use these datasets to make the interesting observation that in submerged biofilms in static liquid cultures, there are two subpopulations of *B. subtilis*: a subpopulation that performs glycolysis and a subpopulation that performs gluconeogenesis. Furthermore, the authors perform spatial-temporal microscopy measurements of cell death in submerged biofilms in such cultures. The authors then summarize the findings of this manuscript and from previous studies in a model of biofilm development for static liquid cultures.

Overall, the authors present an interesting manuscript, which seems to me like a "rough diamond" that would benefit from lots of polishing. Some of this polishing is required for a publication in my opinion, while other polishing is more optional but would substantially improve the impact of the manuscript, as described in my major comments below.

The most important novelty of the manuscript are the two main datasets I mentioned above, which have resource value for the *Bacillus subtilis* research community, and potentially also for the biofilm research community. However, the manuscript does not make full use of these datasets and does not make them readily available to the community in a useful way, as described in my major comments below.

Major comments:

1. The whole manuscript is in need of English language editing. There are many small grammatical mistakes, which makes it difficult for me to be completely certain of what the authors would like to say. This makes it difficult for me to review the accuracy of the content and whether the interpretations and conclusions are supported by the data. I would like to review the manuscript again after such a careful language editing.

2. The title of the manuscript is "Multi-scale transcriptome unveils...", yet the authors do not show transcriptomes at multiple scales. The authors simply measured transcriptomes from different conditions and locations. This dataset is interesting and useful, but it is not "multi-scale". Therefore, the title needs to be changed.

3. The main novelty of the paper is the presentation of two nice datasets, which have significant resource value. However, the actual data is not accessible in a format that it is useful. The GEO accession number gives raw reads and a CSV file that appears to be useless without further information. The authors need to provide at least a human-readable Excel file where the readers can look up interesting genes. Furthermore, given the resource value of this dataset, the authors should also go beyond an Excel file and provide a way to make this data more accessible, perhaps through integration into SubtiWiki or another tool that lets readers view and investigate the dataset.

4. Transcriptomic data in Figure 1 and Figures S1-S5: Although the resource value of this dataset is nice, the analysis and interpretations of this dataset that are presented in the manuscript are underwhelming. The authors do not make much use of this great dataset. Specific comments:

4a. The primary analysis of this dataset was the identification of groups of co-regulated genes. Some of these groups of co-regulated genes contain some genes that belong to one or more functional categories. The results of this analysis are not used later in the manuscript. It is not clear what insights, if any, can be learned from this analysis of the groups of co-regulated genes.

4b. Authors mention in the discussion lines 323-327 that the three biofilm conditions ("mother colony MC", "submerged biofilm SB" and "pellicle PL") have different transcriptome profiles. Pointing this out is fine, but I would expect the authors to reveal if there are any interpretations that can be made from this, beyond simply reporting that there are differences. The paragraph 339-360 points out some interpretations of the transcriptome differences, but it seems

rudimentary and speculative. In my view, the authors should systematically explore and interpret the transcriptomic differences between the different samples.

4c. Can the authors provide an analysis of the different transcriptomic results that provides an insight into the differences between the conditions, and why these differences might occur? Without such an analysis the presentation of this dataset is purely descriptive and without insights.

4d. Currently, the only really meaningful insight that the authors obtain from the transcriptomics results is shown in Figure 3a. Here, the authors observe the counterintuitive upregulation of both *gapA* and *gapB* in the submerged biofilm, which indicates the presence of two subpopulations in these biofilms.

5. Fluorescent reporters and 4D confocal imaging: I think these results are impressive, but there are several issues with the presentation and discussion of this dataset, as described below.

5a. Can the authors please describe in detail how the 17 genes were chosen? The manuscript just states "based on transcriptome data and known gene functions..." (line 153), which is not transparent enough and actually confusing. Why were these 17 genes chosen and not any other genes? Why 17?

5b. The section that describes Figure 2 is titled "Spatio-temporal patterns of gene expression reveals the various heterogeneous subpopulations present during biofilm development". Unfortunately, I cannot see directly how these imaging results reveal heterogeneous subpopulations inside biofilms. What do the authors mean by subpopulations and how do they identify them based on the transcriptional reporters? Each reporter has a different spatio-temporal pattern, but that does not mean that there are 15 different subpopulations.

5c. Lines 169-174: I cannot see how the authors make the conclusions in this paragraph based on the data that they present (Figure 2a and Movie S1). The kymograph shows that around 5 h, there are less cells present, but I cannot conclude more from this data. Movie S1 does not have enough time resolution for me to verify that there are motile cells. For the interpretation that the authors present in this paragraph, additional data must be shown to make this interpretation convincing.

5d. In Figure 2b, the authors present the interesting confocal imaging measurements of 15 transcriptional reporters for submerged biofilms. The description of these results lists which genes are high or low at certain times. It is unclear to me what conclusions or insights the authors draw from these measurements.

6. In the final sentence of the abstract, lines 16-17, the authors mention that the dataset provides insights into dispersal of surface-associated communities, yet the manuscript does not actually present any insights into biofilm dispersal.

7. In my opinion Figure 4 should be merged with Figure 3, as Figure 4 just shows spatio-temporal characterizations of the process already highlighted in Figure 3b.

8. It is not immediately apparent how Figure 5 is connected to the transcriptomics data or the transcriptional reporter data or the glycolysis/gluconeogenesis switch. Therefore, Figure 5 seems isolated and not well integrated into the paper. I understand that Figure 5 is useful for the discussion and the biofilm development model in Figure 6, but the authors should motivate the cell death measurements better in the results section.

9. The manuscript frequently uses abbreviations for the samples (MC, ST, EX, DT, BS, TP, SB, DC, PL), which makes the text difficult to read. The authors should consider spelling out these abbreviations in the whole manuscript – the readers of the manuscript would certainly appreciate this.

Minor comments:

10. It is not clear to me whether the 3 biological replicates were acquired from the same plate/tube/well, or whether they were acquired from independent cultures. This should be

mentioned in the figure caption and methods.

11. In the whole manuscript, the authors use the terminology of "compartments" to describe the different populations throughout the paper. I think this terminology is confusing and inaccurate: the cells acquired from the planktonic culture are not from a different "compartment".

12. Line 171: "separating out" or "spreading out"?

13. In Figure 2b, what is the unit of the color scale – is it simply fluorescence intensity? What are the white spots in the graphs?

14. In Figure S6, the font in the graphs is invisible. What is the y-axis on the bar graphs? Are the color range the same for each image? For each line, it needs to be described what the red, green, white color means.

15. Line 244: Figure 3b does not represent "real-time" spatial monitoring. There is no temporal data in this figure.

16. The title of the results section "Conversion from glycolytic to gluconeogenic regime starts from localised single cell within a glycolytic expressing population", is grammatically flawed and/or not meaningful. I am not sure what "starts from a localized single cell" means here – of course the cells are localized. And of course any change starts from individual cells. I do not understand what the authors mean here.

Reviewer #2 (Remarks to the Author):

In this manuscript Dergham et al explore the existence of heterogeneity of gene expression during biofilm formation using a mesoscale experimental set up. The manuscript relies mostly on RNAseq analysis, and further using some transcriptional fusions and confocal microscopy studies. Overall, the major conclusions are the co-existence of subpopulations and specially of divergent metabolic pathways related to carbon metabolism. I would like to manifest to the authors my major concerns related to the manuscript:

1. This is a very descriptive manuscript, that shows how rich a single species biofilm is in term of specific subpopulations, however, it is my feeling that this study mostly confirms what has been already described in the well-known *Bacillus subtilis* 3610, or even in *Bacillus cereus*: i) that motile and ECM producers are exclusive cell destinies, ii) the level of sporulation in outer layer of the biofilm, iii) the existence of cell death as a step in the process of biofilm or ECM maturation. It is not surprising that we find coexistence of cell types at divergent metabolic stages, and such has been proposed as an ecological strategy exploited by *Bacillus cereus* to adapt to changes in the environmental conditions (NPJ Biofilms Microbiomes. 2020 Jan 15;6:3. doi: 10.1038/s41522-019-0112-7).

2. It would be interesting to have pictures of how biofilms of this strain are formed in liquid culture, and also the swarming phenotype. This would help understanding how the sample were taken, and most importantly, to give relevance to the findings. In this regard, the authors use only one medium, with glucose as a carbon source, have they considered the use of glycerol, in order to see, how the co-existence of metabolically exclusive populations change?

3. In the manuscript it is said, at some point, that a dynamics of gene expression was monitored, however, RNAseq was only done at one time point, most probably when the different cellular compartment was completely assembled. This means that we are only seeing changes that are maintained over time, or even those that only occur at 24 h, when samples were taken. It would be interesting knowing for example, how the different compartment is assembling by monitoring gene expression at different time points. Thus, we will be able to delineate the genetic replacement that dynamically occurs in the process of biofilm formation.

4. There is an important part of the manuscript dedicated to the development of submerged biofilm. This is quite interesting, because it happens in environmental conditions different from what is governing in the air-liquid interphase. The authors show a compilation of gene expressions

of different physiological responses but I hardly find a connection between all of them. Do the authors know what is the oxygen level accessible to cells in the base of the wells, where submerged biofilm are formed? I think it would be interesting knowing if cells in SB are doing aerobic or anaerobic or even fermentation, and that should be compared to pellicle associated cells, and what the author call detached cells. This is at some point described, but not clarified. This information would be valuable to connect cellular metabolic activity with the cellular compartment that is forming at certain time point. What it would be the expected phenotype under growth anaerobic conditions? Would you expect a major tendency to form SB, rather than pellicle?

5. How is the transition from one subpopulation to another, if any, is it similar to what has been described in pellicle for 3610? Cells that are expressing surfactin are also the cells that later are producing the ECM in SB? Or which are the original cells giving rise to sporulation?

6. Maybe I am wrong, but *B. subtilis* 3610 is not forming submerged biofilm, at least in MSgg. Have the authors check whether this phenotype is not reproducible (for 3610) in the medium used in this study? If that is the case, what makes the strain used in this study specially developing such a biofilm compared to 3610? It is a regulated developmental program as described for pellicle and wrinkly colonies, or it is the result of a stochastic process that initiate in stationary phase and continues because of the expression of certain bacterial factors, that otherwise has not been described in this manuscript? Is it possible to abrogate the formation of SB by mutating any of the ECM related genes, or other genes differentially expressed in this compartment, but without affecting pellicle formation or even swarming?

7. The discussion is way too long, and it is my opinion that it should be shorten to put their findings in context to what is known, and what is added to this developmental program. Do the authors think that cell death is necessary for the assembly of SB? What would be the result, if cell death is repressed or at least reduced? Have the author monitor biofilm formation of a *skf* mutant? Apparently, there is a moment when overexpression of this toxin is observed, that overlaps with the expression of competence related genes.

8. With all this information provide in the manuscript, might the authors speculate in the final outcome under different environmental conditions? Are the three cellular compartments always forming, or this is favored by specific nutrient availability or other abiotic conditions? Can these cellular compartments be manipulated by changing carbon or nitrogen sources? which, otherwise, is a situation that may happen in natural settings.

Reviewer #3 (Remarks to the Author):

This project from the groups of Briandet and Hamze explores differential gene expression of *Bacillus subtilis* when grown in several different model biofilm growth configurations (agar colony, pellicle, submerged surface). The paper starts with RNA-seq analysis of population-wide patterns in gene expression in these different sample types. For the submerged biofilms that are more amenable to high resolution imaging, the authors also used a group of 17 fluorescent protein promoter fusion constructs to visualize differential gene expression at a finer scale. A key notable observation from this part of the paper was the sequence and spatial pattern of glycolytic versus gluconeogenic metabolic gene expression.

It is relatively rare to see directed comparison of biofilm-pertinent gene expression in the different model environments often used for this type of research. This main goal of the paper is to be commended in particular, I think. Having noted this, I would say that the paper remains mostly descriptive in its results presentation, and I encountered several issues with the displayed data that would be important to address if this paper were to be revised.

1) One significant issue with the communication in the early part of the paper is the constant reference to the acronym-shortened versions of all the locations biofilms were sampled from. These locations in the various growth conditions (9 in all) are labeled in Figure 1a, but the labeling is not legible at print scale. Clarifying this part of the figure would help a lot here.

2) For the time series data shown in Figure 2 and beyond, I did not see an indication of replication - is it the case that each of the conditions in Figure 2 were only run once for image capture? This

would be problematic from the perspective of reproducibility and not really sufficient for this journal's standards.

3) Related to the point above - many of the subpanels within Figure 2 have some odd patterns. For example, for the *bslA* heat map time series, the time points at which the signal for *bslA* transcription increases (roughly 16 h and 26 h), in addition to the signal within the biofilm increasing, the background signal (the space above the max height of the biofilm) also increases by a similar fold change, by quick reference to the heatmap scaling. This is somewhat concerning as it may mean the apparent increase in signal intensity within the biofilm is actually just an overall increase in total background noise, which can occur for example if the microscope settings were not identical across every time point shown (e.g. change in laser power), or if some other environmental condition changed. Either way, this is one of the reasons the absence of replication is problematic - quantifying these transcriptional change patterns across multiple samples, with estimates of their variance/consistency, would substantially improve the work.

4) It looks as though the heatmaps in Figure 2 are showing projections of total fluorescence averaged across the X and Y dimensions of the viewing field, to show total reporter fluorescence as a function of time and height in the submerged biofilms. It was not clear from the analysis description of the reporter fluorescence is being normalized to the total amount of bacterial volume present. This could be important if the density of cells per unit volume of space in the biofilm is changing over time, which looks often to be the case here.

5) Figure 4 does mention replication - is it correct that the heatmap plots shown are from on representative time series? The spikes and dips in background signal do not appear in these heatmaps, but there is a different odd pattern: At the ~42 h mark, the heatmap signal jumps up and is indicating strong signal at heights up to and probably past the 80 micrometer "ceiling" of the z-stack acquisition. Though the change in reporter expression is clear, there is no change beyond a slight increase in maximum biofilm height is visible in the raw image projections in Figure 4a, though. This latter pattern does not match the heatmap data display for 42h+. This indicates either a problem with the segmentation, or that the way the expression data are normalized per unit bacterial volume has not been described fully.

RESPONSES TO REVIEWER COMMENTS

Reviewer #1 (Remarks to the Author):

This manuscript presents two main datasets: First, a transcriptome dataset from *Bacillus subtilis* in different conditions, including liquid culture, swarms and different types of biofilms. Second, a spatial-temporal microscopy dataset for 17 fluorescent reporters.

The authors use these datasets to make the interesting observation that in submerged biofilms in static liquid cultures, there are two subpopulations of *B. subtilis*: a subpopulation that performs glycolysis and a subpopulation that performs gluconeogenesis. Furthermore, the authors perform spatial-temporal microscopy measurements of cell death in submerged biofilms in such cultures. The authors then summarize the findings of this manuscript and from previous studies in a model of biofilm development for static liquid cultures.

Overall, the authors present an interesting manuscript, which seems to me like a “rough diamond” that would benefit from lots of polishing. Some of this polishing is required for a publication in my opinion, while other polishing is more optional but would substantially improve the impact of the manuscript, as described in my major comments below.

The most important novelty of the manuscript are the two main datasets I mentioned above, which have resource value for the *Bacillus subtilis* research community, and potentially also for the biofilm research community. However, the manuscript does not make full use of these datasets and does not make them readily available to the community in a useful way, as described in my major comments below.

Major comments:

1. The whole manuscript is in need of English language editing. There are many small grammatical mistakes, which makes it difficult for me to be completely certain of what the authors would like to say. This makes it difficult for me to review the accuracy of the content and whether the interpretations and conclusions are supported by the data. I would like to review the manuscript again after such a careful language editing.

To address this concern comprehensively, we have utilised the language editing service provided by Nature Communications. A proficient editor, fluent in English, provided by Springer Nature Authors Services (SNAS C131-F89A-F251-02AC-C33P), meticulously edited the article. The text was carefully edited with the goal of rectifying all grammatical mistakes and enhancing the overall clarity of the manuscript.

2. The title of the manuscript is “Multi-scale transcriptome unveils...”, yet the authors do not show transcriptomes at multiple scales. The authors simply measured transcriptomes from

different conditions and locations. This dataset is interesting and useful, but it is not “multi-scale”. Therefore, the title needs to be changed.

To respond to this comment and better represent the content of our article, we have made the necessary changes to the title. The revised title now reads, "Unravelling mosaic gene expression in diverse *Bacillus subtilis* biofilm communities: Insights from transcriptomics and live-cell imaging". This new title ensures a more precise representation of the research conducted and aligns with the core focus of our investigation.

3. The main novelty of the paper is the presentation of two nice datasets, which have significant resource value. However, the actual data is not accessible in a format that it is useful. The GEO accession number gives raw reads and a CSV file that appears to be useless without further information. The authors need to provide at least a human-readable Excel file where the readers can look up interesting genes. Furthermore, given the resource value of this dataset, the authors should also go beyond an Excel file and provide a way to make this data more accessible, perhaps through integration into SubtiWiki or another tool that lets readers view and investigate the dataset.

Taking into account the factors addressed above, we've developed an interactive website that facilitates the exploration of condition-dependent expression profiles with remarkable single-nucleotide precision. This website has been constructed using Genoscapist, providing a gateway to access these profiles via https://genoscapist.migale.inrae.fr/seb_bsubbiofilm/. Furthermore, we have included two supplementary Excel files: S1, which contains the comprehensive RNA-Seq data, and S2, which encompasses the results of the gene set enrichment analysis.

4. Transcriptomic data in Figure 1 and Figures S1-S5: Although the resource value of this dataset is nice, the analysis and interpretations of this dataset that are presented in the manuscript are underwhelming. The authors do not make much use of this great dataset. Specific comments:

4a. The primary analysis of this dataset was the identification of groups of co-regulated genes. Some of these groups of co-regulated genes contain some genes that belong to one or more functional categories. The results of this analysis are not used later in the manuscript. It is not clear what insights, if any, can be learned from this analysis of the groups of co-regulated genes.

4b. Authors mention in the discussion lines 323-327 that the three biofilm conditions (“mother colony MC”, “submerged biofilm SB” and “pellicle PL”) have different transcriptome profiles. Pointing this out is fine, but I would expect the authors to reveal if there are any interpretations that can be made from this, beyond simply reporting that there are differences. The paragraph 339-360 points out some interpretations of the transcriptome differences, but it seems

rudimentary and speculative. In my view, the authors should systematically explore and interpret the transcriptomic differences between the different samples.

4c. Can the authors provide an analysis of the different transcriptomic results that provides an insight into the differences between the conditions, and why these differences might occur? Without such an analysis the presentation of this dataset is purely descriptive and without insights.

4d. Currently, the only really meaningful insight that the authors obtain from the transcriptomics results is shown in Figure 3a. Here, the authors observe the counterintuitive upregulation of both *gapA* and *gapB* in the submerged biofilm, which indicates the presence of two subpopulations in these biofilms.

The questions addressed above appear to primarily revolve around the superficial description of the transcriptome data. In this revised new version, we have made efforts to go deeper into the analysis and interpretation of this extensive dataset rather than staying at a descriptive level. Consequently, several modifications and idea organisations have been implemented:

- We developed a Genoscapist interactive website and included Excel tables S1 and S2 in the new version, which greatly facilitated the process of tracking and interpreting analysis of the RNA-seq data. A substantial portion of this analysis has been incorporated into the results section, spanning from lines 124 to 241 in the new version, with their supplementary figures S2-S4.
- We incorporated a PCA analysis into Figure 1c, in the new version, which allowed a more distinct visualisation of the closeness or separation among the distinct populations. Moreover, a Venn diagram, presented in supplementary Figure S2 of the new version, effectively illustrates the interplay between sets of DEGs within the biofilm populations in relation to both the stationary and exponential phases.
- In the discussion section, our aim was to enrich the interpretation of the data and to be straight forward. We achieved this by combining the transcriptome information with live-imaging data related to the reported genes/operons. Within this context, we devoted sections to the exploration of motility, matrix production, anaerobic respiration, sporulation, and dead cells. Additionally, we provided a more comprehensive understanding of submerged biofilm development over time.

5. Fluorescent reporters and 4D confocal imaging: I think these results are impressive, but there are several issues with the presentation and discussion of this dataset, as described below.

5a. Can the authors please describe in detail how the 17 genes were chosen? The manuscript just states “based on transcriptome data and known gene functions...” (line 153), which is not transparent enough and actually confusing. Why were these 17 genes chosen and not any other genes? Why 17?

Regarding the selection of the 17 genes, we have revised the corresponding section in the manuscript to provide a more detailed explanation. The new version now reads as follows:

For the *cggR-gapA* and *gapB* genes we have mentioned our interest in their selection in lines 263-265, new version “*Interestingly, the glycolytic cggR-gapA operon and the gluconeogenic gapB/pckA genes were strictly oppositely regulated in all the selected populations, except for the submerged biofilm, in which both were upregulated (Fig. 2a).*”

As for the selection of other genes the choice was based on different interest mentioned in lines 324-336, new version “*Based on the transcriptome data, the submerged biofilm exhibited downregulation of major matrix genes compared to the other biofilm populations (Supplementary Table S1, Fig. 1d). This suggests that these genes were either consistently expressed at a low level within the submerged biofilm or were expressed only during a brief period before or after the 24-hour transcriptome time point. To further investigate the characteristics of the submerged biofilm, we opted to observe these particular genes and other genes involved in various biological functions that could potentially influence biofilm development; these chosen genes were distributed among different clusters within the global heatmap (Fig. 1d, Supplementary Fig. S9). Using fluorescent reporter transcriptional fusions, we closely monitored the expression of these genes/operons involved in matrix synthesis (*epsA-O*, *tapA-tasA*, *bslA*, *srfAA-AD*, *ypqP*, *capA-E*), motility (*hag*), exoprotease synthesis (*aprE*), carbon metabolism (*ackA*), competence (*comGA-GG*), cannibalism (*skfA*), respiration (*ctaA*, *narG-I*), and sporulation (*spoIIIGA-sigG*, *spoVC-VT*).”*

In addition to better describing the diversity of expression within this subset of genes, a new heatmap presenting the levels of expression for each of those genes under different conditions is presented in Supplementary Figure S9b.

5b. The section that describes Figure 2 is titled “Spatio-temporal patterns of gene expression reveals the various heterogeneous subpopulations present during biofilm development”. Unfortunately, I cannot see directly how these imaging results reveal heterogeneous subpopulations inside biofilms. What do the authors mean by subpopulations and how do they identify them based on the transcriptional reporters? Each reporter has a different spatio-temporal pattern, but that does not mean that there are 15 different subpopulations.

To clarify, we have replaced the term "subpopulations" with "cell types" throughout the relevant sections of the manuscript, except when referring to the contrasting set of cells under opposing metabolic conditions (glycolytic vs. gluconeogenic subpopulations). We acknowledge that the original terminology might have caused confusion.

This section has undergone a transformation from a highly descriptive passage to a more concentrated one, in which we were able to visualise the coordination of the expression of the selected reported genes. This allowed us to conclude: “*These observations revealed that gene*

expression in submerged biofilms is not uniform but rather exhibits dynamic spatiotemporal mosaic patterns.” lines 366-367, new version.

5c. Lines 169-174: I cannot see how the authors make the conclusions in this paragraph based on the data that they present (Figure 2a and Movie S1). The kymograph shows that around 5 h, there are less cells present, but I cannot conclude more from this data. Movie S1 does not have enough time resolution for me to verify that there are motile cells. For the interpretation that the authors present in this paragraph, additional data must be shown to make this interpretation convincing.

The particular biological process had been documented in a similar configuration but with a different growth medium in our previous publication (ref: 36 in the new version). And since we have observed similar expression profiles of matrix and motility genes in this synthetic B-medium, this has prompted us to state that sessile cells differentiate to motile ones.

However, it is accurate to acknowledge that the supplementary Movie S5 in the new version lacks the requisite temporal resolution. Consequently, we are unable to accentuate the phenomenon outlined in the previous version. For this reason, this section has been modified accordingly in lines 308-313 in the new version to focus mainly on the occurrence of waves of dead cells “*Bacteria adhered to the surface and formed chains of sessile cells during the first hours of incubation, and then, between approximately 15 and 24 hours, clusters of dead cells were observed above the formed biofilm (Fig. 2d, Supplementary movie S5). After this first wave, the density of dead cells decreased, with a slight increase in the live population until approximately 42 hours, when a second wave of dead cells was observed (Fig. 2d, Supplementary movie S5)*”. In addition, we have added a movie (supplementary movie S7, new version) showing the motility of cells through the expression of *hag* during submerged biofilm development.

5d. In Figure 2b, the authors present the interesting confocal imaging measurements of 15 transcriptional reporters for submerged biofilms. The description of these results lists which genes are high or low at certain times. It is unclear to me what conclusions or insights the authors draw from these measurements.

In the revised version of the article, we have made changes to the presentation of this figure (Figure 3b, new version) to provide more comprehensive insights into the data. The figure now appears at the end of the article (lines 368), serving as an opening illustration to showcase the diversity of spatiotemporal patterns of gene expression within submerged biofilms.

The description of the results pertaining to the 15 transcriptional reporters has been expanded to offer a more in-depth analysis of the observed gene expression patterns. Specifically, we now elaborate on the significance of genes being expressed at different levels and specific time points during submerged biofilm development (lines 337-367 in the new version). This

expanded analysis aims to provide a clearer understanding of the biological implications and potential regulatory roles of these genes within the submerged biofilm context.

6. In the final sentence of the abstract, lines 16-17, the authors mention that the dataset provides insights into dispersal of surface-associated communities, yet the manuscript does not actually present any insights into biofilm dispersal.

In response to this comment, we have promptly removed the mention of dispersion from the final sentence of the abstract. We agree that the manuscript does not explicitly present data or findings related to biofilm dispersal.

7. In my opinion Figure 4 should be merged with Figure 3, as Figure 4 just shows spatio-temporal characterizations of the process already highlighted in Figure 3b.

In the updated version, Figure 2a, 2b, and 2c represent the combined figures from the previous version, which were originally designated as Figures 3 and 4.

8. It is not immediately apparent how Figure 5 is connected to the transcriptomics data or the transcriptional reporter data or the glycolysis/gluconeogenesis switch. Therefore, Figure 5 seems isolated and not well integrated into the paper. I understand that Figure 5 is useful for the discussion and the biofilm development model in Figure 6, but the authors should motivate the cell death measurements better in the results section.

In the updated version, Figure 2d was originally Figure 5b in the previous version. In this version, our intention was to establish a correlation between the reestablishment of the glycolytic regime and the simultaneous onset of cell death. To strengthen our argument, we meticulously examined the metabolic regimen by tracking the expression of glycolytic/gluconeogenic genes under varying carbon sources (Supplementary Fig. S8; lines 314-320 in the new version). Moreover, we investigated the effect of *skfA* mutation on the occurrence of cell death, revealing its impact on the first wave (Supplementary Fig. S12, S16, and S17; lines 359-361 / 434-442).

9. The manuscript frequently uses abbreviations for the samples (MC, ST, EX, DT, BS, TP, SB, DC, PL), which makes the text difficult to read. The authors should consider spelling out these abbreviations in the whole manuscript – the readers of the manuscript would certainly appreciate this.

In the new version, these abbreviations were expanded in the main text.

Minor comments:

10. It is not clear to me whether the 3 biological replicates were acquired from the same plate/tube/well, or whether they were acquired from independent cultures. This should be mentioned in the figure caption and methods.

To address this concern, we want to confirm that the three biological replicates were indeed acquired from independent cultures. This information has been clarified in both the figure caption and the methods section of the revised article.

11. In the whole manuscript, the authors use the terminology of “compartments” to describe the different populations throughout the paper. I think this terminology is confusing and inaccurate: the cells acquired from the planktonic culture are not from a different “compartment”.

To address this concern, we have made the necessary changes throughout the entire manuscript. The term "compartment" has been replaced with the more appropriate and accurate term "population". This modification ensures a clear and consistent representation of the different cell groups observed in our study, including those acquired from the planktonic cultures. By adopting the term "population," we aim to avoid confusion and accurately describe the diverse cell communities investigated in our research.

12. Line 171: “separating out” or “spreading out”?

Aligned with the new modifications, this aspect has been removed.

13. In Figure 2b, what is the unit of the color scale – is it simply fluorescence intensity? What are the white spots in the graphs?

In the new version, Figure 3b was originally Figure 2b in the previous version. The unit of the colour scale in this Figure is indeed fluorescence intensity, expressed in arbitrary units (a.u.). We have now included this information in the figure and its caption to provide a clearer understanding of the data representation.

Regarding the white spots in the graphs, they indicate intensity values below a threshold limit, representing no significant signal for that specific tile, which was added in the legend of the corresponding Figure 3b.

14. In Figure S6, the font in the graphs is invisible. What is the y-axis on the bar graphs? Are the color range the same for each image? For each line, it needs to be described what the red, green, white color means.

To address these concerns, the y-axis on the bar graphs in Figure S6 (Supplementary Fig. S15, new version) represents the intensity of gene expression due to the transcriptional fusions to a fluorescent reporter gene. However, we apologise for the oversight in the previous version, where the font size made the graphs difficult to read. In response to this, we have increased the font size of the graphs in the revised version of the supplementary material to improve their visibility.

Regarding the colour range in the images, it indeed represents the intensity of gene expression, with each colour corresponding to different levels of expression. However, we understand that the description of the colour representation was lacking in the previous version. Therefore, in the revised figure caption, we have included a detailed description of what the red, green, and grey colours signify to provide clear and comprehensive information “*The expression of the gene is visualised using confocal imaging, with Gfp fluorescence indicated in green and mCherry fluorescence shown in red. The grey contrast was achieved either through chemical staining (using Syto9 or Syto61) or by expressing a second reporter gene fusion in strains containing such a fusion*”.

15. Line 244: Figure 3b does not represent “real-time” spatial monitoring. There is no temporal data in this figure.

Aligned with the modifications, this aspect has been changed.

16. The title of the results section “Conversion from glycolytic to gluconeogenic regime starts from localised single cell within a glycolytic expressing population”, is grammatically flawed and/or not meaningful. I am not sure what “starts from a localized single cell” means here – of course the cells are localized. And of course any change starts from individual cells. I do not understand what the authors mean here.

In response to this feedback, we have thoroughly revised the entire paragraph to ensure clarity and coherence with the merged Figures 3 and 4 (Figure 2a, 2b and 2c in the new version). The new title of the results section now reads: “*Spatial transcriptomics detected mutually exclusive carbon metabolic regimes occurring within a biofilm.*” lines 246-247 in the new version.

Reviewer #2 (Remarks to the Author):

In this manuscript Dergham et al explore the existence of heterogeneity of gene expression during biofilm formation using a mesoscale experimental set up. The manuscript relies mostly on RNAseq analysis, and further using some transcriptional fusions and confocal microscopy studies. Overall, the major conclusions are the co-existence of subpopulations and specially of divergent metabolic pathways related to carbon metabolism. I would like to manifest to the authors my major concerns related to the manuscript:

1. This is a very descriptive manuscript, that shows how rich a single species biofilm is in term of specific subpopulations, however, it is my feeling that this study mostly confirms what has been already described in the well-known *Bacillus subtilis* 3610, or even in *Bacillus cereus*: i) that motile and ECM producers are exclusive cell destinies, ii) the level of sporulation in outer layer of the biofilm, iii) the existence of cell death as a step in the process of biofilm or ECM maturation. It is not surprising that we find coexistence of cell types at divergent metabolic stages, and such has been proposed as an ecological strategy exploited by *Bacillus cereus* to adapt to changes in the environmental conditions (NPJ Biofilms Microbiomes. 2020 Jan 15;6:3. doi: 10.1038/s41522-019-0112-7).

We appreciate the reviewer's feedback and acknowledge that our manuscript primarily offers a descriptive study of a single species biofilm, which may echo some findings previously reported in *Bacillus subtilis* NCIB3610 and *Bacillus cereus*. Indeed, we have also verified that the NCIB3610 had a similar phenotypic behaviour in the synthetic B-medium used for this study (lines 386-389, new version; Supplementary movie S10; Supplementary Fig. S14).

However, we believe that our study possesses distinctive characteristics that contribute to its originality and significance:

-Comparative Analysis: Our work compares *Bacillus* biofilm models using the same strain, growth media, and protocol. This comparative approach enables us to assess colony biofilm, submerged biofilm, and floating biofilm at the transcriptomic level simultaneously. To our knowledge, this is the first study to offer such a comprehensive comparison.

-Local Bacterial Cell Expression: We utilise RNAseq on locally captured populations and fluorescent reporter systems to decipher bacterial cell expression within highly heterogeneous systems. This approach provides valuable insights into the dynamics and diversity of gene expression at a localised level, offering a deeper understanding of biofilm biology.

-Exploration of Unknown Genes: We have identified specific subpopulations that exhibit significantly high overexpression of genes with unknown functions. This discovery opens doors to new functional studies, potentially uncovering novel regulatory mechanisms and biological processes in biofilm development and behaviour.

-Data Accessibility: We have made significant datasets available to the scientific community. These include the RNAseq data accessible through user-friendly tools like Genoscapist, facilitating exploration of different genes across nine distinct conditions. Additionally, we provide a robust 4D gene expression images dataset that can be reanalyzed, for example using advanced AI-assisted image tool solutions.

We believe that these unique aspects contribute to the scientific value and novelty of our research, further enriching the understanding of biofilm complexity and behaviour in *Bacillus subtilis*.

2. It would be interesting to have pictures of how biofilms of this strain are formed in liquid culture, and also the swarming phenotype. This would help understanding how the sample were taken, and most importantly, to give relevance to the findings. In this regard, the authors use only one medium, with glucose as a carbon source, have they considered the use of glycerol, in order to see, how the co-existence of metabolically exclusive populations change?

In response to the reviewer's suggestion, a new Supplementary Figure S14 presents pictures of the *Bacillus subtilis* NDmed in the different biofilm models. In addition, Supplementary movies S1, S2 and S3 present the kinetics of the swarming process. These images provide visual context to understand how the biofilms are formed in static liquid and the swarming models, enhancing the relevance of the choice of the different selected populations.

Furthermore, novel experiments were performed to address the utilisation of different carbon sources; we now provide data on biofilms grown with either glucose, glycerol, or malate. These experiments were conducted to explore how the coexistence of metabolically exclusive subpopulations may vary depending on the different carbon sources used. The results of these experiments are now presented in a new Supplementary Figure S8.

In the revised version of the manuscript, we have mentioned the inclusion of these additional experiments and data on lines 314-321.

3. In the manuscript it is said, at some point, that a dynamics of gene expression was monitored, however, RNAseq was only done at one time point, most probably when the different cellular compartment was completely assembled. This means that we are only seeing changes that are maintained over time, or even those that only occur at 24 h, when samples were taken. It would be interesting knowing for example, how the different compartment is assembling by monitoring gene expression at different time points. Thus, we will be able to delineate the genetic replacement that dynamically occurs in the process of biofilm formation.

While it is true that in the current manuscript, RNAseq was performed at a single time point, we want to highlight that we previously conducted a global time-course transcriptomic profiling analysis during biofilm formation in a static liquid model (ref: 36 in the revised

version). In this new contribution, our experimental strategy was designed to combine local whole-genome RNAseq data with the selection of a list of genes of interest. For these specific genes, we constructed fluorescent reporter systems to monitor their local expression with high temporal resolution. This approach allows us to capture the dynamics of gene expression on a transcriptomic timescale, with steps observed at intervals of less than 1 hour over a period of 48 hours. By employing this novel approach, we can delineate the genetic replacement and dynamic changes occurring during the process of submerged biofilm formation at a much finer timescale than previously reported. The combination of whole-genome RNAseq and high-resolution temporal monitoring through fluorescent reporters provides unprecedented insights into the temporal dynamics of gene expression during biofilm development. In addition, Supplementary movies S1, S2 and S3 present the kinetics during the swarming development, showing, for example, the differential expression of motility and matrix genes involved in the process.

4. There is an important part of the manuscript dedicated to the development of submerged biofilm. This is quite interesting, because it happens in environmental conditions different from what is governing in the air-liquid interphase. The authors show a compilation of gene expressions of different physiological responses but I hardly find a connection between all of them. Do the authors know what is the oxygen level accessible to cells in the base of the wells, where submerged biofilm are formed? I think it would be interesting knowing if cells in SB are doing aerobic or anaerobic or even fermentation, and that should be compared to pellicle associated cells, and what the author call detached cells. This is at some point described, but not clarified. This information would be valuable to connect cellular metabolic activity with the cellular compartment that is forming at certain time point. What it would be the expected phenotype under growth anaerobic conditions? Would you expect a major tendency to form SB, rather than pellicle?

Regarding the oxygen levels accessible to cells in the base of the wells where submerged biofilms are formed, we performed new experiments to gain a better understanding. We utilised a fluorescent-based commercial solution that allows us to monitor oxygen concentration over time in the microtiter plate wells. The results from three replicates (Supplementary Fig. S11c) confirmed our previous observation that the oxygen level inside the wells drops below the detection limit after approximately 4 hours of *Bacillus subtilis* growth (ref: 36 in the revised version). This corresponds to the initiation of the overexpression of genes involved in anaerobic metabolism, such as *narG* (Supplementary Fig. S11a and S11b).

The newly acquired data and observations have been incorporated into the revised manuscript (lines 346-349 and 410-412), providing valuable insights into the cellular metabolic activity of biofilm development under specific oxygen conditions.

Besides, we conducted experiments under microaerophilic conditions (Figure below). The outcomes indicated only slight variations in biofilm features among the different biofilm

models. However, definitive conclusions could not be drawn due to the fact that the conditions employed were not entirely anaerobic.

[This figure presents a comparative study of *B. subtilis* NDmed cultivated under either aerobic or microaerophilic conditions (using AnaeroGen, Oxoid Ltd, England) for the various biofilm models.]

5. How is the transition from one subpopulation to another, if any, is it similar to what has been described in pellicle for 3610? Cells that are expressing surfactin are also the cells that later are producing the ECM in SB? Or which are the original cells giving rise to sporulation?

Indeed, understanding the transition from one cell type to another in our biofilm models is a fascinating aspect to explore. However, our current experimental design is not optimised to address this specific question at the single-cell resolution required for detailed analysis. To achieve single-cell dynamics and track the fate of individual cells expressing surfactin or later producing the extracellular matrix in submerged biofilms, we would need to develop a more complex and diluted system. For example, incorporating a low proportion of cells expressing fluorescent fusions in a non fluorescent WT population. While this approach is potentially promising and interesting, it would require significant experimental development and optimization, and it is beyond the scope of our current study.

6. Maybe I am wrong, but *B. subtilis* 3610 is not forming submerged biofilm, at least in MSgg. Have the authors check whether this phenotype is not reproducible (for 3610) in the medium used in this study? If that is the case, what makes the strain used in this study specially developing such a biofilm compared to 3610? It is a regulated developmental program as described for pellicle and wrinkly colonies, or it is the result of a stochastic process that initiate in stationary phase and continues because of the expression of certain bacterial factors, that otherwise has not been described in this manuscript? Is it possible to abrogate the formation of SB by mutating any of the ECM related genes, or other genes differentially expressed in this compartment, but without affecting pellicle formation or even swarming?

To address the reviewer's concerns, we performed experiments to compare the phenotypes of *B. subtilis* NDmed and *B. subtilis* NCIB3610 in the specific medium used in this study. Both strains were capable of forming the three types of biofilms, including submerged biofilms. We observed some variations, particularly in the floating pellicle, but overall, both strains displayed similar biofilm-forming abilities (Supplementary movie S10; Supplementary Fig. S14).

Regarding the specific formation of submerged biofilms in the strain used in this study, we agree with the reviewer's observations. In contrast to swarming and floating pellicle formation, we did not identify a specific gene whose mutation would abrogate submerged biofilm formation in this strain (ref: 7 in the revised version). Based on current knowledge, *B. subtilis* biofilm models could be classified into two categories: those involving a regulated developmental program (such as pellicle and swarming) and those that result from stochastic processes (like submerged biofilms and colony biofilms). Furthermore, the particular factors and regulatory mechanisms governing the development of submerged biofilms may not only differ from those of other biofilm types, but they could also vary based on environmental conditions and potentially vary between different strains.

7. The discussion is way too long, and it is my opinion that it should be shortened to put their findings in context to what is known, and what is added to this developmental program. Do the authors think that cell death is necessary for the assembly of SB? What would be the result, if cell death is repressed or at least reduced? Have the author monitor biofilm formation of a *skf* mutant? Apparently, there is a moment when overexpression of this toxin is observed, that overlaps with the expression of competence related genes.

In response to the reviewer's comment, we have revised and shortened the discussion section to provide a more concise and focused overview of our findings and their implications in the context of existing knowledge.

The extracellular DNA (eDNA) liberated from dead cells has been described as a major determinant of biofilm rheological properties. Regarding the role of cell death in the assembly of biofilm, we performed additional experiments as suggested. We generated a *skfA* mutant of *B. subtilis* NDmed-GFP strain. The *skfA* mutation significantly affected colony morphology, leading to the loss of 3D structuration and wrinkles (Supplementary Fig. S17), which have been

associated with cell death. In the submerged biofilm model, we generated spatiotemporal kymographs to investigate the effect of *skfA* mutation on the occurrence of cell death. This strongly suggested the involvement of SkfA in the occurrence of the first wave of cell death (lines 357-361 and 434-445 in the revised version; Supplementary Fig. S12, S16, and S17).

8. With all this information provide in the manuscript, might the authors speculate in the final outcome under different environmental conditions? Are the three cellular compartments always forming, or this is favored by specific nutrient availability or other abiotic conditions? Can these cellular compartments be manipulated by changing carbon or nitrogen sources? which, otherwise, is a situation that may happen in natural settings.

Recognizing the potential variations in biofilm formation that could arise from alterations in growth medium or other environmental factors, the primary aim of this investigation was to contrast the spatial transcriptomic profiles among distinct biofilm types, all employing an identical *B. subtilis* strain and growth medium.

Using the synthetic B-medium with different carbon sources (glucose, glycerol, or malate) allowed us to observe either similarities or differences in the biofilm formation process (figure below). For instance, bacteria cultured in glycerol showed a slightly reduced submerged biofilm but stronger pellicle. While in malate the submerged biofilm was not affected contrary to the formation of the pellicle that remains discontinuous and swarming was aborted.

[This figure presents a comparative study of *B. subtilis* NDmed cultivated under either glycolytic (glucose or glycerol) or gluconeogenic (malate) conditions for the various biofilm models.]

Reviewer #3 (Remarks to the Author):

This project from the groups of Briandet and Hamze explores differential gene expression of *Bacillus subtilis* when grown in several different model biofilm growth configurations (agar colony, pellicle, submerged surface). The paper starts with RNA-seq analysis of population-wide patterns in gene expression in these different sample types. For the submerged biofilms that are more amenable to high resolution imaging, the authors also used a group of 17 fluorescent protein promoter fusion constructs to visualize differential gene expression at a finer scale. A key notable observation from this part of the paper was the sequence and spatial pattern of glycolytic versus gluconeogenic metabolic gene expression.

It is relatively rare to see directed comparison of biofilm-pertinent gene expression in the different model environments often used for this type of research. This main goal of the paper is to be commended in particular, I think. Having noted this, I would say that the paper remains mostly descriptive in its results presentation, and I encountered several issues with the displayed data that would be important to address if this paper were to be revised.

1) One significant issue with the communication in the early part of the paper is the constant reference to the acronym-shortened versions of all the locations biofilms were sampled from. These locations in the various growth conditions (9 in all) are labeled in Figure 1a, but the labeling is not legible at print scale. Clarifying this part of the figure would help a lot here.

In response to the issue with the constant reference to acronym-shortened versions of the sampled biofilm locations, we have increased the font size in Figure 1a to enhance legibility at the print scale.

2) For the time series data shown in Figure 2 and beyond, I did not see an indication of replication - is it the case that each of the conditions in Figure 2 were only run once for image capture? This would be problematic from the perspective of reproducibility and not really sufficient for this journal's standards.

We want to assure that each of the conditions depicted in Figure 3 in the new version and beyond was indeed replicated at least three times independently to ensure the reliability and reproducibility of our findings. To address these concerns and adhere to the journal's standards, we have made the necessary revisions. The three replicates for each kymograph shown in Figure 3 are now presented in a supplementary material (Supplementary Fig. S13), providing a more comprehensive view of the experimental results. Furthermore, to enhance transparency and facilitate reproducibility, we made the raw data used for generating these kymographs accessible in an online repository (<https://doi.org/10.57745/Z511A6>). This will allow interested readers and researchers to access the original data and verify our findings independently.

3) Related to the point above - many of the subpanels within Figure 2 have some odd patterns. For example, for the *bslA* heat map time series, the time points at which the signal for *bslA* transcription increases (roughly 16 h and 26 h), in addition to the signal within the biofilm increasing, the background signal (the space above the max height of the biofilm) also increases by a similar fold change, by quick reference to the heatmap scaling. This is somewhat concerning as it may mean the apparent increase in signal intensity within the biofilm is actually just an overall increase in total background noise, which can occur for example if the microscope settings were not identical across every time point shown (e.g. change in laser power), or if some other environmental condition changed. Either way, this is one of the reasons the absence of replication is problematic - quantifying these transcriptional change patterns across multiple samples, with estimates of their variance/consistency, would substantially improve the work.

We thoroughly investigated the potential factors that could lead to the odd patterns observed in some subpanels of Figure 3 in the new version, particularly in the *bslA* heatmap time series. To address the possibility of changes in microscope settings, specifically the laser power, affecting the results, we checked the stability of the laser power over the entire 48-hour experimental period. We ensure that the observed changes in signal intensity are not essentially due to fluctuations in laser power. This information has been added in the revised version “*The intensity of a calibrated reflected laser remained consistent within an acceptable range (not more than 10% variation) throughout the duration of the experiment*” lines 634-636 in the new version.

Regarding the light band observed in the kymographs of *bslA*, that indicates this phenomenon is related to the dispersion of free motile fluorescent cells in the well. This observation aligns with what we observed in the 4D confocal movies (Supplementary movies S6 and S7) where the presence of these motile cells became apparent.

4) It looks as though the heatmaps in Figure 2 are showing projections of total fluorescence averaged across the X and Y dimensions of the viewing field, to show total reporter fluorescence as a function of time and height in the submerged biofilms. It was not clear from the analysis description of the reporter fluorescence is being normalized to the total amount of bacterial volume present. This could be important if the density of cells per unit volume of space in the biofilm is changing over time, which looks often to be the case here.

This is correct; the heatmaps in Figure 3 in the new version show projections of total fluorescence, which are averaged across the X and Y dimensions of the viewing field. This representation allows us to visualise the total reporter fluorescence as a function of time and height in the submerged biofilms. Regarding the analysis description, we want to clarify that the fluorescent signal reported in the heatmaps is indeed the intensity of fluorescence, measured in arbitrary units. We acknowledge that normalisation to the total amount of bacterial volume present within the biofilm field could be relevant, particularly if the density of cells per unit volume of space in the biofilm changes over time, as is often observed in our experiments. In

the current analysis, we did not explicitly normalise the fluorescent signal to the biofilm biovolume, mainly due to the challenge of extracting accurate biofilm biomass information from the acquired cell field in a time course experiment. However, to address this limitation, we have included a kymograph reporting the fluorescence signal of a strain constitutively reporting the GFP (*Phyp-gfp*). This reference kymograph allows the reader to correct the fluorescent signal for the biofilm biomass, providing a more comprehensive understanding of the dynamics of the reporter fluorescence in relation to the biofilm structure and growth over time.

5) Figure 4 does mention replication - is it correct that the heatmap plots shown are from on representative time series? The spikes and dips in background signal do not appear in these heatmaps, but there is a different odd pattern: At the ~42 h mark, the heatmap signal jumps up and is indicating strong signal at heights up to and probably past the 80 micrometer "ceiling" of the z-stack acquisition. Though the change in reporter expression is clear, there is no change beyond a slight increase in maximum biofilm height is visible in the raw image projections in Figure 4a, though. This latter pattern does not match the heatmap data display for 42h+. This indicates either a problem with the segmentation, or that the way the expression data are normalized per unit bacterial volume has not been described fully.

The Figure 2c in the new version (Figure 4 in the previous version) is one representative of a time series. The experiment presented was reproduced a minimum of three separate times (raw data used are accessible in an online repository (<https://doi.org/10.57745/Z511A6>)) from which three kymographs were extracted and presented in the Supplementary Figure S13. This duplication was carried out independently to establish the credibility and replicability of our results. We acknowledge that the kymographs did not accurately represent the raw image due to an inadequate and narrow scale, resulting in a strong fluorescent signal appearing overly intense. To address this, we have expanded the scale range to more effectively depict the confocal image.

Reviewer #1 (Remarks to the Author):

The authors have addressed all of my concerns very well, and made changes that improved the paper significantly. This paper is now insightful and accurate.

Reviewer #2 (Remarks to the Author):

In this new version of the manuscript, the authors have added new information as suggested by reviewers, which I do believe has improved clarity and quality of the data presented. Specifically, I do acknowledge the effort the authors have employed to provide with satisfactory answers to most of my comments. I do not have more specific comments to ask.

Reviewer #3 (Remarks to the Author):

The updates to the manuscript in response to referee comments are much appreciated. My main reaction to this paper is not all that different from the first round, though. This is primarily a descriptive report that presents more as a technical addendum to a prior publication from these groups, and it does not in my opinion quite reach the high standards of this journal in terms of mechanistic depth and broad interest. If the other referees together feel differently, however, I am happy to defer to them.